# Understanding aerosol microphysical properties from 10 years of data collected at Cabo Verde based on an unsupervised machine learning classification

Xianda Gong[1], Heike Wex[1], Thomas Müller[1], Silvia Henning[1], Jens Voigtländer[1], Alfred Wiedensohler[1], and Frank Stratmann[1]

[1]Department Experimental Aerosol and Cloud Microphysics, Leibniz-Institute for Tropospheric Research (TROPOS), Leipzig, 04318, Germany

**Correspondence:** Xianda Gong (gong@tropos.de)

**Abstract.** The Cape Verde Atmospheric Observatory (CVAO), which is influenced by both, marine and desert dust air masses, has been used for long-term measurements of different properties of the atmospheric aerosol from 2008 to 2017. These properties include particle number size distributions (PNSD), light absorbing carbon (LAC) and concentrations of cloud condensation nuclei (CCN) together with their hygroscopicity. Here we summarize the results obtained for these properties and use an unsupervised machine learning algorithm for the classification of aerosol types. Five types of aerosols, i.e., marine, freshly-formed, mixture, moderate dust and heavy dust, were classified. Air masses during marine periods are from the Atlantic Ocean and during dust periods are from the Sahara. Heavy dust was more frequently present during wintertime, whereas the clean marine periods were more frequently present during springtime. It was observed that during the dust periods CCN number concentrations at a supersaturation of 0.30% are roughly 2.5 times higher than during marine periods, but the hygroscopicity ($\kappa$) of particles in the size range from $\sim$30 to $\sim$175 nm during marine and dust periods are comparable. The long-term data presented here, together with the aerosol classification, can be used as a base to improve our understanding of annual cycles of the atmospheric aerosol in the eastern tropical Atlantic and on aerosol-cloud interactions and it can be used as a base for driving, evaluating and constraining atmospheric model simulations.

## 1 Introduction

Aerosol particles are known to impact the cloud formation, life cycle of clouds and global radiative forcing. These impacts are closely related to aerosol particle physical and chemical properties. Atmospheric aerosol particles originate from a vast variety of anthropogenic and natural sources (Pöschl, 2005). In this study, the focus is on naturally occurring aerosol types. Natural aerosol sources include primary mineral dust, sea spray, and wildfire aerosol, as well as precursors for new particle formation, such as sulfur- and carbon-containing gases from vegetation, ocean environment, and volcanoes, that can subsequently form particles in the atmosphere (Carslaw et al., 2010).

Mineral dust is likely the most abundant aerosol type by mass in the atmosphere (Kok et al., 2017), with estimates for the dust emission ranging from 1000-2150 Tg yr$^{-1}$ (Zender et al., 2004). The direct dust–climate feedback is likely in the range

of -0.04 to +0.02 Wm$^{-2}$ K$^{-1}$, and it may account for a substantial fraction of the total aerosol feedback in the climate system (Kok et al., 2018). Furthermore, mineral dust can change cloud properties, i.e., serve as cloud condensation nuclei (CCN) (Karydis et al., 2011) or ice-nucleating particles (INPs) (Sassen et al., 2003; DeMott et al., 2003; Kanji et al., 2017). Based on a global transport model, Karydis et al. (2011) predicted that the annual average contribution of insoluble mineral dust to total

particle number concentrations is up to 15% over South American and Australian deserts and up to 40% in the North African and Asian deserts.

Wind-generated particles on the surface of the ocean are one of the most important constituents of atmospheric aerosol particles (Prather et al., 2013; Modini et al., 2015; Quinn et al., 2015). Generally, aerosol particles present in marine environments are referred to as marine particles. These particles comprise ocean emitted primary sea spray aerosol which includes particles

containing organic components and sea salt, with high fractions of the former in smaller particles and vice versa, together with newly formed particles originating from precursor gases such as dimethylsulphide (DMS), long-range transport of dust, anthropogenic pollution and biomass burning. Marine aerosol in the size range below 10 $\mu$m diameter usually featured a tri-modal size distribution, showing that the production of marine aerosols is based on different mechanisms (Modini et al., 2015; Wex et al., 2016; Brooks and Thornton, 2018). Marine aerosol particles' hygroscopicity and their ability to act as CCN can be

controlled by marine ocean processes such as biological activity and wind-dependent sea spray generation (Quinn et al., 2015). For example, in the North Atlantic Ocean, O'Dowd et al. (2004) found that the organic fraction dominated the sub-micron aerosol mass and contributed 63% (45% water-insoluble and 18% water-soluble) during algae bloom periods, while this value decreased to 15% during the lowest ocean activity periods. Sea spray aerosol (SSA, including directly emitted organic matter and inorganic salt) contributed less than 30% of the CCN population on a global basis, with the exception of high latitudes of

the Southern Ocean where it may be larger (Quinn et al., 2017). Marine aerosols are expected to be a source of INPs, but less efficient than dust and biological particles (Wilson et al., 2015; DeMott et al., 2016; Gong et al., 2020a).

The Cabo Verde (a.k.a. Cape Verde) islands are located in the tropical northern Atlantic Ocean. In 2006, the Cape Verde Atmospheric Observatory (CVAO) was set up by British, German and Cabo Verde scientific institutes. CVAO provides a platform for studying remote marine aerosols, as well as long-range transported continental and Saharan-dust aerosols (Fomba

et al., 2013; van Pinxteren et al., 2020). CVAO is located at the far edge of the island in the direction of air mass inflow to the island. Therefore, air masses observed at CVAO are free from local anthropogenic pollution. The aerosol composition at CVAO varies spatially and temporally together with meteorological conditions. Generally, CVAO receives clean marine air masses from the northern Atlantic Ocean during late spring and summer, while it receives air masses with high dust load from the Sahara during late fall and winter (Fomba et al., 2014). Gong et al. (2020b) characterized different aerosol types (marine,

mixture, and moderate dust) at CVAO, for data collected during September and October 2017.

In this study, a detailed overview of long-term particle microphysical data sets obtained at CVAO is provided. With this, we extend the characterization by Gong et al. (2020b) by including additional information on seasonal variations, and provide additional data on the CCN population and particle hygroscopicity for marine and dust aerosol dominated environments. Particle number size distributions (PNSDs) in the size range from 20 nm to 10 $\mu$m and light absorbing carbon (LAC) mass

concentration were characterized from April 2008 to December 2017. CCN number concentration ($N_{CCN}$) and particle hygro-

scopicity (expressed as $\kappa$) were characterized from October 2015 to March 2016 and from September to November 2017. The corresponding backward trajectories were calculated from the HYbrid Single-Particle Lagrangian Integrated Trajectory (HYS-PLIT) Model from April 2008 to December 2017. It is known that short-term measurements are not representative because the real world has large spatial heterogeneity and temporal variability. The long-term measurement at CVAO provided precious

particle microphysical data sets to help understand the particles' nature in a dust-marine environment. Furthermore, long-term measurements can be used for a better constrain of modeling efforts (Reddington et al., 2017).

It is challenging to analyze and interpret the large data sets resulting from long-term measurements. Machine learning algorithms can build powerful models that can quickly and automatically analyze bigger, more complex data sets and deliver faster, accurate results than traditional statistical methods. The machine learning algorithms include supervised and unsupervised

approaches, depending on using labeled or unlabeled data to help predict outcomes. In general, supervised machine learning requires upfront human intervention based on prior knowledge on a connection between input data and desired output values. By feeding data and desired outputs into an algorithm, weights are adjusted in the model until the output is met sufficiently well, and the model can then be used for characterizing further input data. On the other hand, unsupervised machine learning algorithms learn the inherent structure of the data without using explicitly provided labels. The goal is to infer the natural

structure present within a set of data points.

The choice of unsupervised or supervised machine learning algorithms will depend on the scientific questions to be answered. Each method has its advantages and disadvantages. The different machine learning algorithms can be very sensitive to user-defined parameters and implementation. A number of freely available open-source and easy-to-use packages in popular programming languages, such as Scikit-learn in Python, have been developed in academia and industry. Machine learning

approaches have been increasingly used for particle classification. For example, Christopoulos et al. (2018) implemented a supervised decision tree for aerosol classification for on-line single-particle mass spectrometry. Atwood et al. (2019) classified aerosol population types using an unsupervised K-means cluster analysis. In the present study, an unsupervised machine learning algorithm K-mean is used to help us understand better the particle microphysical properties at CVAO.

## 2 Experiment and Methods

### 2.1 Sampling location and measurement setup

The measurement station (CVAO, 16°51′49″ N, 24°52′02″ W) is located at the northeastern shore of the São Vicente island. It is about 10 m a.s.l and 70 m from the coastline. The São Vicente island itself is in the north Atlantic Ocean, $\sim$900 km off the African coast. This region constantly experiences northeasterly winds, with an average wind speed of about 7.3 m s$^{-1}$ (Carpenter et al., 2010; Fomba et al., 2014). The average annual temperature at São Vicente is about 23.6 $\pm$ 4.0 °C. Cabo

Verde has very little precipitation, with an annual rainfall of about 3 to 10 events and 24-350 mm, mainly between August and October (Fomba et al., 2013).

An aerosol PM$_{10}$ inlet, installed on top of a 32 m tower, was deployed to remove particles larger than 10 $\mu$m. The purpose of installing an aerosol inlet on such a high tower is to minimize the influence of sea spray aerosol generated in the surf zone

**Table 1.** Measured parameters, the respective instrumentations, and measurement periods.

| Parameter | Instrument | Measurement range | Time resolution | Sampling period |
|---|---|---|---|---|
| PNSD | SMPS & APS | 20 nm to 10 $\mu$m | 1h mean | April 2008 to December 2017 [1] |
| LAC | MAAP | - | 1h mean | April 2008 to December 2017 [1] |
| $N_{CCN}$ | CCNC | SS=0.15, 0.20, 0.30, 0.50, and/or 0.70% | 1h (each SS) [2] | October 2015 to March 2016 & September to November 2017 |
| $\kappa$ | CCNC & PNSD | SS=0.15, 0.20, 0.30, 0.50, and/or 0.70% | 1h (each SS) [2] | October 2015 to March 2016 & September to November 2017 |
| Backward trajectory | HYSPLIT | - | 1h | April 2008 to December 2017 |

[1] For SMPS, APS, and MAAP measurements, there were occasional gaps due to instrumental failures.

[2] Data from roughly 6.5 to 9.5 mins of measurements contributed to the hourly averages.

at the coastline. Downstream of the $PM_{10}$ inlet, a vertical stainless-steel sampling pipe (1/2 in. outer diameter), together with a 1-meter long diffusion dryer, was placed on top of a measurement container. All microphysical instruments were placed inside this air-conditioned container. The microphysical instruments included a TROPOS-type MPSS (mobility particle size spectrometer), an APS (aerodynamic particle sizer, model 3321, TSI Inc.), a MAAP (multi-angle absorption photometer, type 5012, Thermo Scientific Inc.) and a CCNC (cloud condensation nuclei counter, Droplet Measurement Technologies). More details concerning the CVAO station setup can be found in Gong et al. (2020b). The measured parameters, instruments and measurement periods are summarized in Tab. 1.

## 2.2 Particle size

PNSDs in the size range from 20 nm to 10 $\mu$m were measured by using a TROPOS-type MPSS and an APS. Both MPSS and APS were monitored remotely and calibrated regularly during the measurements. The necessary parameters controlling the sizing of the instruments were logged continuously. Obvious problems were solved by a station engineer. Calibration of sensors took place irregularly but at least once a year and the sizing was checked using traceable PSL particles of sizes 200, 1000, and 2000 nm. Sizing errors are typically below 5%. A traceable check of the counting efficiency is not possible at the station. Instead, instruments calibrated at TROPOS (in the WCCAP - World Calibration Center for Aerosol Physics) were sent to the station infrequently for inter-comparison measurements or, if needed, for replacing instruments.

The electrical mobility distribution (measured by the MPSS in the size range from 20 to ∼800 nm) is converted to a PNSD by applying an inversion algorithm to correct for multiply charged aerosol particles (Wiedensohler, 1988) and diffusional losses (Wiedensohler et al., 2012). In this study, APS data, i.e., PNSDs for sizes above ∼700 nm, was involved to enable a multiple charge correction of the MPSS data when applying the inversion algorithm (for more detailed explanations see Pfeifer et al., 2014). Prior to using APS data in the inversion, the aerodynamic diameters as measured by APS had to be converted to geometric diameters, for which a dry dynamic shape factor and particle density are needed. The dry dynamic shape factor ($\chi$)

and density ($\rho$) of sodium chloride are 1.08 and 2160 kg m$^{-3}$ (Kelly and McMurry, 1992; Gysel et al., 2002), respectively, whereas the dynamic shape factor and density of mineral dust are 1.25 (Kaaden et al., 2009) and within a range of 2450-2700 kg m$^{-3}$ (Haywood et al., 2001). Averaged shape factor and density of 1.17 and 2000 kg m$^{-3}$ were used in this study (Schladitz et al., 2011). Overall, the combined PNSD is given based on volume equivalent particle diameters in this study. Size-dependent particle losses within the inlet were corrected (refer to Appendix A). Based on the combined PNSDs, total particle number concentration ($N_{total}$) and particle number concentrations (PNC) in different size ranges can be calculated.

## 2.3 Light absorbing carbon

Light absorbing carbon (LAC) concentrations were derived from measurements with a multi-angle absorption photometer (MAAP, Petzold and Schönlinner, 2004). The MAAP measures the optical transmission and reflection at a wavelength of 637 nm of a filter while it is loaded with particles. An internal algorithm calculates an equivalent black carbon concentration (eBC, Petzold et al., 2013) using a mass specific absorption coefficient (MAC) of 6.6 m$^2$g$^{-1}$. The MAC is a conversion factor between the light absorption coefficient ($\sigma_{abs}$), which is the primary measured parameter, and a corresponding eBC, which are related by:

$$\sigma_{abs} = eBC * MAC \tag{1}$$

In Müller et al. (2011), it was shown that the values for eBC and $\sigma_{abs}$ had to be corrected by up to a factor of 1.05 to correct for an originally wrong assumption of the wavelength. Furthermore, filter-based absorption photometers are affected by an artifact due to light scattering particles. While the multi-angle approach of the MAAP compensates for this to a large extent, it was shown in Müller et al. (2011) that $\sigma_{abs}$ is still overestimated by about 1% of the value of the light scattering coefficient ($\sigma_{sca}$) at the wavelength of 637 nm. The calculation of the fully corrected light absorption coefficient ($\sigma_{abs,corr}$) is shown in the following equation:

$$\sigma_{abs,corr} = 1.05 * \sigma_{abs} - 0.01 * \sigma_{sca} \tag{2}$$

Since the $\sigma_{sca}$ was not measured directly, it was derived from PNSDs using Mie scattering theory (Bohren and Huffman, 2008). The required complex refractive index was assumed to be 1.49-0.01*i. This value is a reasonable assumption for less absorbing marine aerosols (Hess et al., 1998). However, due to the assumption of a refractive index and the assumption of spherical particles, the quality of calculated scattering coefficients are not sufficiently good, e.g., for the use in radiative transfer calculations. Therefore, scattering data are not presented in this study. The calculated scattering coefficients serve as the best estimate for minimizing artifacts in the absorption measurements. A comparison of probability density functions (PDFs) between $\sigma_{abs}$ and $\sigma_{abs,corr}$ is shown in the Appendix B. It can be seen later in the study (Sect. 3.2.1.), that further improvements for the scattering corrections are of importance, especially for high single-scattering albedos, since Eq. 2 is merely a first order correction. A deeper investigation requires instruments not affected by scattering artifacts, e.g. photo-acoustic photometers.

## 2.4 Cloud condensation nuclei and particle hygroscopicity

A cloud condensation nuclei counter (CCNC, Roberts and Nenes, 2005) was used to measure CCN number concentration ($N_{CCN}$). The main part of the CCNC is a cylindrical continuous-flow thermal-gradient diffusion chamber. A constant streamwise temperature gradient is established to adjust a quasi constant centerline supersaturation. During the measurement, the sampled aerosol particles are guided through this chamber within a sheath flow along the centerline. Depending on the supersaturation adjusted in the CCNC and the particle's hygroscopic properties, particles can become activated to droplets.

In this study, the calibrated supersaturations were 0.15%, 0.20%, 0.30%, 0.50% and/or 0.70%. The supersaturation was switched every 12 or 15 minutes for five or four subsequently used supersaturations, respectively. Therefore, the $N_{CCN}$ values at each supersaturation have a 1h resolution. The measurement data during the first 5 minutes and the last 30 seconds at each supersaturation were excluded from the data analysis to assure a stable column temperature when switching the supersaturation, and the remaining data points were averaged. To ensure the accuracy of the results, a supersaturation calibration was done repeatedly during the long-term deployment of the instrument at the site (more details concerning calibration method in Gysel and Stratmann, 2013).

Whether or not a particle can be activated to a droplet depends on its chemical composition, its dry size and the maximum supersaturation it encounters (according to Köhler theory, Köhler, 1936). A single parameter $\kappa$ can be used to describe the relationship between particle hygroscopicity and dry diameter at which the particle will be activated (Petters and Kreidenweis, 2007). Assuming the surface tension of the examined solution droplets ($\sigma_{s/\alpha}$) to be that of pure water, in this study $\kappa$ was calculated as follows:

$$\kappa = \frac{4A^3}{27d_{crit}^3 \ln^2(1+SS)} \tag{3}$$

with

$$A = \frac{4\sigma_{s/\alpha}M_\omega}{RT\rho_\omega} \tag{4}$$

where SS and $d_{crit}$ are supersaturation in % and the critical diameter above which all particles activate into droplets at this given supersaturation, respectively. $M_\omega$ is the molar mass of water; $R$=8.314 J (K mol)$^{-1}$ is the ideal gas constant; $T$ is the absolute temperature; $\rho_\omega$ is the density of water.

In this method, aerosol particles are assumed to be internally-mixed. All particles in the neighborhood of a given size are assumed to have a similar $\kappa$. A specific particle can be activated to a droplet if its dry diameter is larger than or equal to $d_{crit}$ at a fixed supersaturation. Therefore, $d_{crit}$ is the diameter at which the cumulative particle number concentration from the largest diameter to $d_{crit}$ is equal to $N_{CCN}$. In this study, the measured $N_{CCN}$ and PNSDs are both used to derive $d_{crit}$.

## 2.5 Backward trajectories

A backward trajectory calculation was conducted with the HYbrid Single-Particle Lagrangian Integrated Trajectory (HYSPLIT) Model (Rolph, 2003; Stein et al., 2015). HYSPLIT calculations were based on GDAS (Global Data Assimilation Sys-

tem) meteorological datasets with a 1 degree spatial resolution. The backward trajectories were calculated for 6 days with a resolution of 6 hours, arriving at 200 m above the measurement site.

## 2.6 Unsupervised machine learning algorithm

An unsupervised K-means algorithm was used in this study to classify aerosol particle types. The K-means model clusters samples by trying to separate data into $k$ groups of equal variance, minimizing a criterion known as the inertia or within-cluster sum-of-squares. Given a set of observations ($x_1$, $x_2$, ..., $x_n$), where each observation is a d-dimensional real vector, K-means clustering aims to partition the $n$ observations into $k$ ($\leq n$) sets $C = C_1$, $C_2$, ..., $C_k$ so as to minimize the within-cluster sum-of-squares. Formally, the objective is to find:

$$\sum_{i=0}^{n} \min_{u_j \in C} (||x_i - u_j||^2) \tag{5}$$

where $u_j$ is the mean of points in $C_j$.

It is important to start the analysis by looking into the number of clusters, $k$, to specify the optimal $k$. In this study, we used an elbow plot and Calinski-Harabasz score (Caliński and Harabasz, 1974) to determine the optimal $k$, which is explained in more detail in Appendix C.

## 3 Results and discussion

### 3.1 Overview of data sets

#### 3.1.1 Time series of particle number size distribution

Particle number size distribution (PNSD) is one of the most important features of atmospheric aerosols, as particles in different size ranges have different formation routes and behaviors. The contour plots for PNSDs of the supermicron and submicron particle are shown in Fig. 1 (a) and (b), respectively. The color scale indicates particle number concentration (dN/dlogD$_p$) in number per cubic centimeter. Because of a relatively lower number concentration of supermicron particles, different color bar scales were adopted for submicron and supermicron particles in Fig. 1. Most of the time, Aitken and accumulation modes in the submicron size range and a coarse mode in the supermicron size range are observed. The Aitken mode ranges from ∼20 to ∼80 nm, and the accumulation mode ranges from ∼80 to ∼1000 nm. The number concentration of supermicron particles shows a high variation, from 1 to above 100 cm$^{-3}$, with a median value of 3.8 cm$^{-3}$.

#### 3.1.2 Seasonal variation of light absorbing carbon

The monthly variation of $\sigma_{abs}$ and $\sigma_{abs,corr}$, averaged over all years, is shown in Fig. 2. For $\sigma_{abs,corr}$, values were mostly below $0.5\,\mathrm{Mm}^{-1}$. However, higher values of up to $1\,\mathrm{Mm}^{-1}$ were observed in October, December and January. The light absorbing aerosols usually include black carbon, brown carbon and dust. The CVAO station is free of local anthropogenic

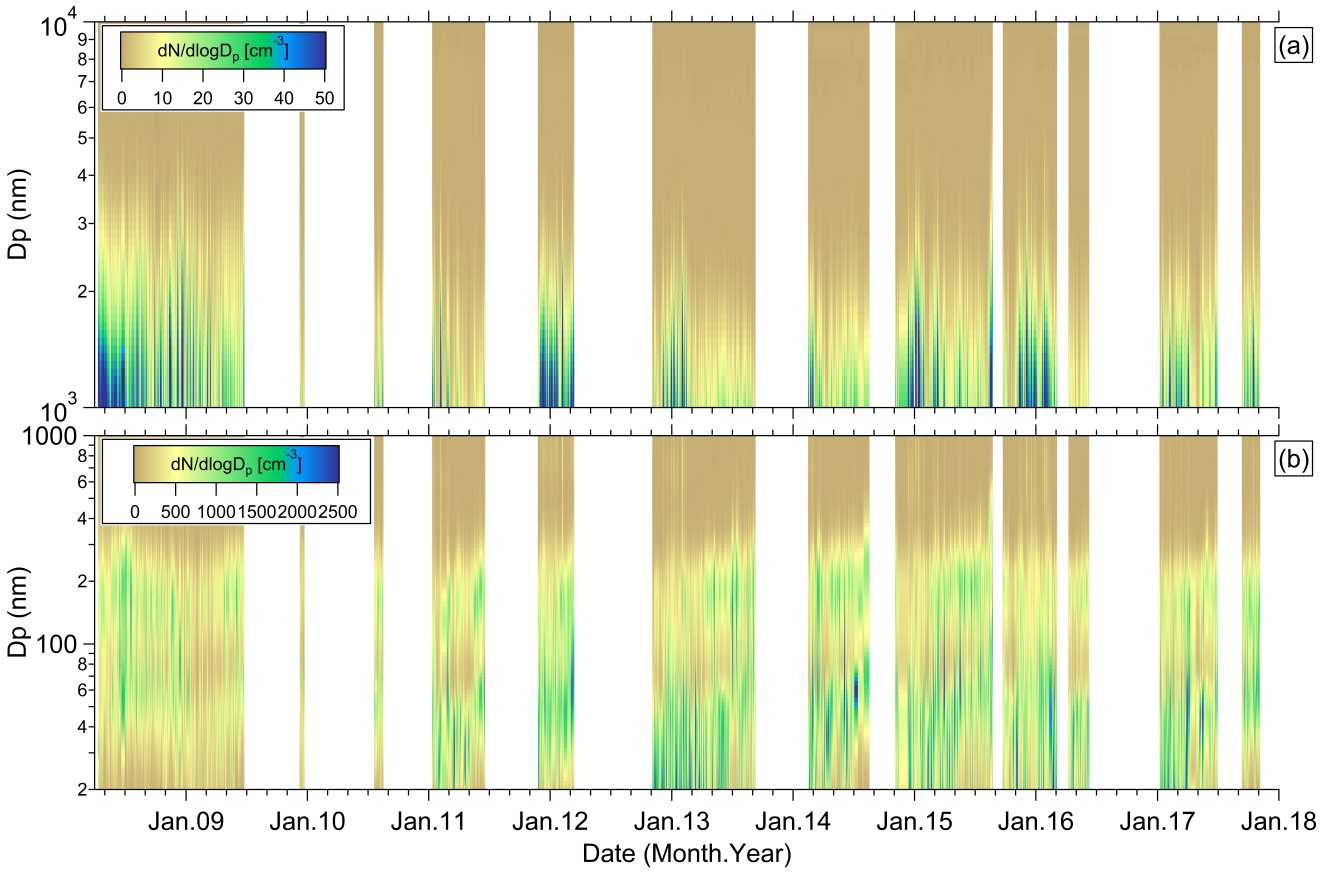

**Figure 1.** Contour plots of hourly averaged PNSDs in the supermicron size range (1000 nm to 10 $\mu$m) in panel (a) and the submicron size range (20 to 1000 nm) in panel (b).

pollution because it is located at the northeastern shore and experiences constant northeasterly winds. Therefore it is highly unlikely that black carbon and brown carbon alone could explain the observed variation of the $\sigma_{abs}$ values. The higher values of $\sigma_{abs}$ coincide with months during which high dust concentrations were generally observed in this study. The variation of LAC during different aerosol types is discussed at the end of Sect. 3.2.1.

### 3.1.3  Time series and analysis of CCN

Figure 3 shows the time series of CCN number concentration ($N_{\mathrm{CCN}}$) in the upper panel, $d_{\mathrm{crit}}$ in the middle panel and $\kappa$ values in the lower panel, with different colors for different supersaturations. $N_{\mathrm{CCN}}$ shows large variability. $N_{\mathrm{CCN}}$ at a supersaturation of 0.30 ($N_{\mathrm{CCN,\,0.30\%}}$) varied from $\sim$10 to above 1000, with a mean value of 336 cm$^{-3}$. The high variability of $N_{\mathrm{CCN}}$ also can be seen in the wide-spread probability density function of CCN number concentrations (PDF-$N_{\mathrm{CCN}}$), shown in Fig. 4 (a). The PDF-$N_{\mathrm{CCN}}$ shows multiple modes at supersaturations of 0.15%, 0.20%, 0.30%, 0.50% and 0.70%, which indicates that

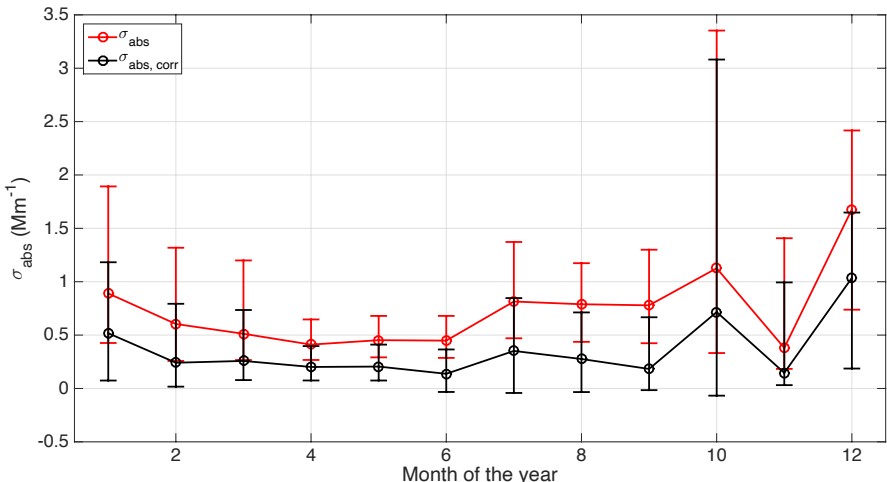

**Figure 2.** Monthly variation of the LAC values. Circles are median values. Error bars are 25th and 75th percentiles.

particles acting as CCN arriving at CVAO originate from different sources. The boxplot of $N_{CCN}$ at different supersaturations is shown in Fig. 5 (a). It is clear that $N_{CCN}$ values increase towards a higher supersaturation, because increasingly smaller particles can be activated to droplets at a increasingly higher supersaturations. The mean/median values of $N_{CCN}$ also exhibit a large variability, increasing from a mean of 233 to 485 $cm^{-3}$ for a variation in supersaturations from 0.15% to 0.70%.

Time series of $d_{crit}$ values at all supersaturations (0.15%, 0.20%, 0.30%, 0.50% and 0.70%) are shown in Fig. 3 (b). The mean value of $d_{crit}$ decreases from 131 nm at a supersaturation of 0.15% to 40 nm at a supersaturation of 0.70% (Fig. 5). As can be seen from Fig. 4, for the lowest supersaturation of 0.15%, $d_{crit}$ is above the size of 100 nm, located in the accumulation mode, whereas for the supersaturations of 0.70% and 0.50%, $d_{crit}$ is below the size of 70 nm, located in the Aitken mode. The derived $\kappa$ at a given supersaturation represents the particle averaged hygroscopicity around the corresponding $d_{crit}$. Therefore,

hygroscopicities derived at supersaturations of 0.70% and 0.50% can be assumed to be representative of the Aitken, whereas hygroscopicities derived at a supersaturation of 0.15% can be assumed to be representative of the accumulation mode. $d_{crit}$ at supersaturations of 0.20% and 0.30% are from about 70 to 100 nm. In this size range, marine aerosol particle size distributions typically show a minimum between Aitken and accumulation mode (Modini et al., 2015; Gong et al., 2020b), also known as Hoppel minimum (Hoppel et al., 1986). The hygroscopicity derived at supersaturations of 0.20% to 0.30% represents a mixture

of particles from both the accumulation and Aitken mode.

Figure 3(c) shows the time series of the particle hygroscopicity parameter $\kappa$. The $\kappa$ values at each supersaturation show a larger variability during the whole campaign compared to a previous study of a one-month intensive measurement campaign at CVAO (Gong et al., 2020b), with one standard deviation varying from 0.10 (SS=0.15%) to 0.16 (SS=0.20%, 0.30%). This also can be seen in the wide-spread PDF of $\kappa$ at different supersaturations (Fig. 4 (c)). The PDF-$\kappa$ values at each supersaturation

show uni-model distributions, except for the supersaturation of 0.70%. The PDF-$\kappa$ at a supersaturation of 0.70% is bimodal featuring a small mode with a peak at 0.16 and a large mode with a peak at 0.36. At this supersaturation of 0.70%, the larger

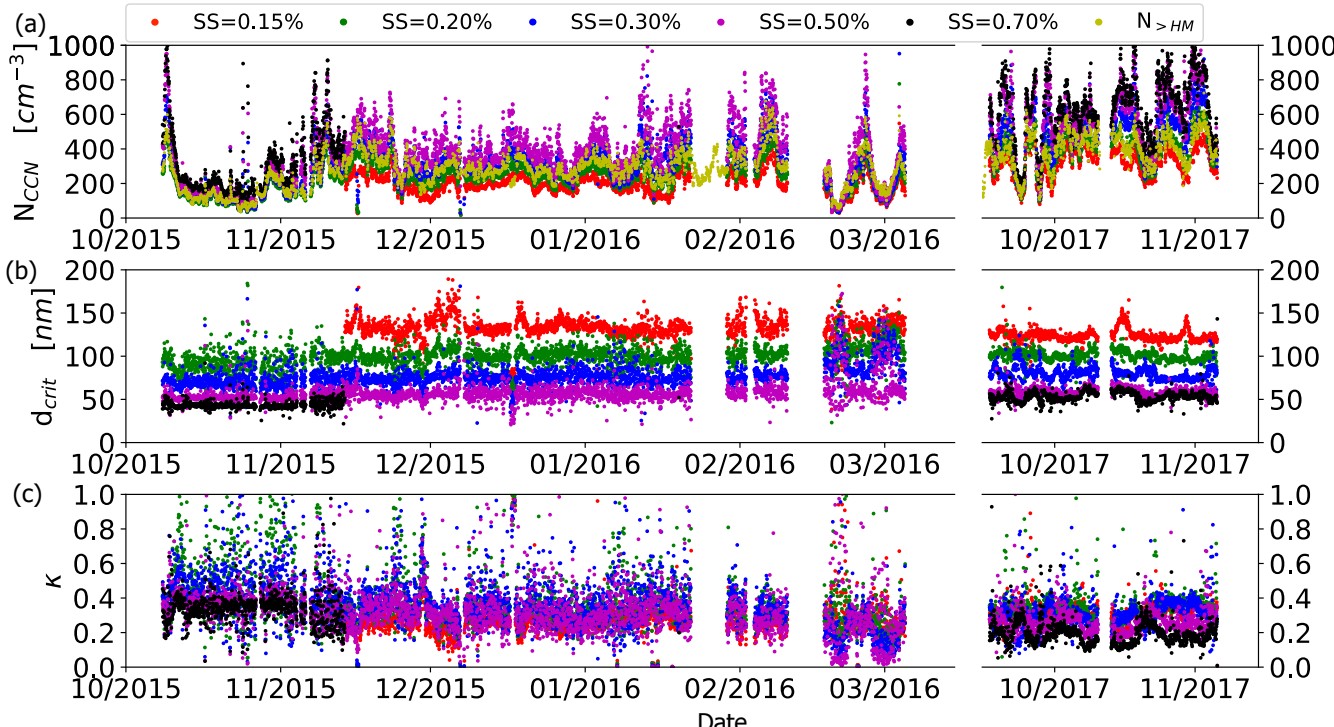

**Figure 3.** Time series of $N_{CCN}$ in panel (a), $d_{crit}$ in panel (b) and $\kappa$ in panel (c) with 1h resolution at each supersaturation. Red, green, blue, purple and black dots indicate supersaturations (SS) of 0.15%, 0.20%, 0.30%, 0.50% and 0.70%, respectively. Yellow dots in the upper panel show the time series of particle number concentration for particles larger than the Hoppel minimum diameter ($N_{>HM}$).

$\kappa$ values are present when CCN number concentration is low, and vice versa. Figure 5 (c) shows the overall $\kappa$ as a function of supersaturation. A slightly increasing trend of $\kappa$ was observed from supersaturations 0.15% to 0.20%, whereas a slightly decreasing trend of $\kappa$ value was observed from 0.20% and above. However, there is no strongly pronounced difference between $\kappa$ at different supersaturations.

5     Figure 5 (d) shows $\kappa$ as a function of $d_{crit}$ and plotted error bars indicate standard and geometric standard deviations for $d_{crit}$ and $\kappa$, respectively. In the size range from 50 to 100 nm (Aitken mode), a slightly increasing trend of $\kappa$ over increasing $d_{crit}$ is observed, indicating that the sulfate produced from DMS oxidation rather than organics might be a major species for particle condensational growth. In the accumulation mode, $\kappa$ is getting smaller. But the overall uncertainty in these derived values is rather large so that no further interpretation of the $\kappa$ values at different sizes will be given. The overall average $\kappa$

10  value is 0.32. As this value was derived for different supersaturations and hence different particle sizes, it has limited use, only. However, in Sect. 3.3, we will discuss $\kappa$ values in more detail, related to most extremely differing air masses. $\kappa$ values did not vary much between these air masses nor between different particle sizes, so that the here given average value may be of some use to characterize the aerosol at least roughly. Still, we want to clearly point out that no data exists for the summer months,

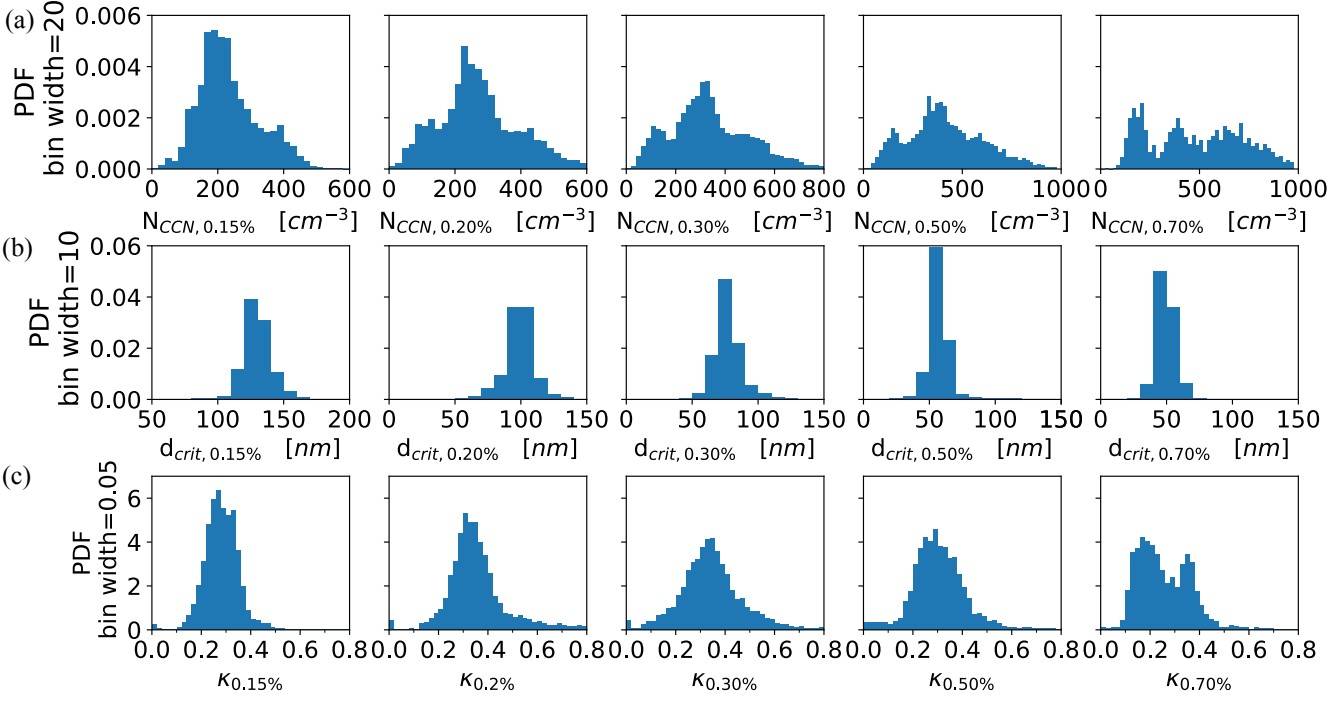

**Figure 4.** Probability of density functions (PDFs) of $N_{CCN}$ (a), $d_{crit}$ (b) and $\kappa$ (c) at supersaturations from 0.15% to 0.70%, based on all available data.

further limiting the applicability of this average value. In Barbados (the easternmost island in the Caribbean), both modeling (Pringle et al., 2010) and measurements (Wex et al., 2016) suggested that $\kappa$ of marine aerosol in the tropical Atlantic can show an annual variation with lowest values of around 0.3 to 0.4 in the summer months. Contrary, for Cabo Verde, Pringle et al. (2010) suggested values for $\kappa$ of up to 0.6 for the time spanning April until July, while they suggest lower values of roughly
0.3 for fall and winter months, in agreement with our measurements.

The Hoppel minimum divides particles that were not activated to a cloud droplet, yet, from other, larger ones, which were. Therefore, the supersaturation of 0.30% can be seen to roughly represent the averaged maximum supersaturations in the clouds along the path of the sampled air masses at the CVAO during our measurement periods. On the other hand, we can use the integrated particle number concentration for particle sizes above the Hoppel minimum ($N_{>HM}$) to estimate atmospheric CCN
number concentrations at a supersaturation of 0.30% at CVAO, as shown in Fig. 3 upper panel, where a good agreement between measured $N_{CCN,0.30\%}$ and calculated $N_{>HM}$ values can be seen. When comparing these values in a scatter plot, the coefficient of determination (R) is 0.89 (Fig. D1 in Appendix D), with high correlations also observed when regarding data on a monthly basis (Fig. D2 in Appendix D). By this, the CCN number concentration can be extended from the 9 months during which $N_{CCN}$ was measured with a CCNC to a range of 10 years for which $N_{>HM}$ values can be derived based on measured
PNSDs. If the Hoppel minimum is not present, which means during the moderate and heavy dust periods (discussed in Sect.

3.2.1), we used the cumulative particle number concentration above 100 nm to estimate CCN number concentration. The 100 nm was chosen because the $d_{\text{crit}}$ was about 100 nm at a supersaturation of 0.30% during dust periods (discussed in Sect. 3.3), and it also coincides with a boundary between two modes in PNSD of the moderate dust.

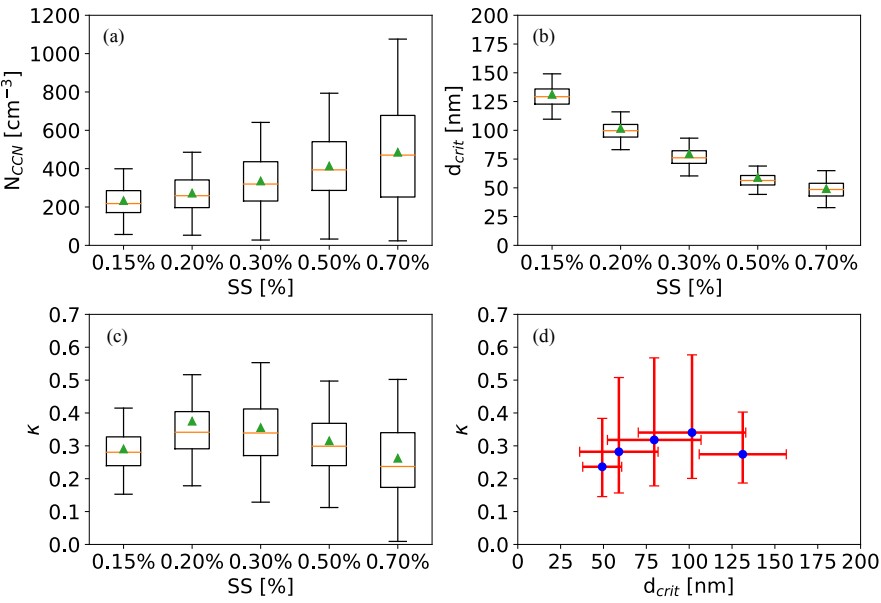

**Figure 5.** Boxplots of $N_{\text{CCN}}$ (a), $d_{\text{crit}}$ (b) and $\kappa$ (c) at supersaturations (SS) from 0.15% to 0.70% based on all available data. Whiskers show the 90th to 10th percentiles. Green solid triangles show the mean values. (d) $\kappa$ as a function of $d_{\text{crit}}$. Error bars of $d_{\text{crit}}$ show 1 standard deviation. Error bars of $\kappa$ show 1 geometric standard deviation.

### 3.1.4 Air mass analysis

To understand the air mass origin and transport, we analyzed the backward trajectories at CVAO. Carpenter et al. (2010) analyzed radiosonde data from October 2007 to August 2008 and found that marine boundary layer (MBL) height varied from ~300 to 1500 m at Cabo Verde. From on-site balloon measurements, Gong et al. (2020b) found that the MBL was typically well mixed, with boundary layer heights from about 550 to 1100 m. An occasionally decoupled layer at around 400 to 600 m was also observed during September and October 2017. Therefore, we used backward trajectories starting at 200 m altitude

to be sure to represent aerosol collected at CVAO. During air mass transport, the aerosol in an air mass might be altered due to precipitation events. We disregarded backward trajectories from our analysis (~4.73% of total backward trajectories) when total precipitation exceeded 20 mm in the past 6 days.

Figure 6 shows the relative frequency of backward trajectories over the 10-year measurements. Most of the backward trajectories featured paths over the North Atlantic Ocean and the coastal area in Africa and arrived at CVAO from the northwest

direction. A subgroup of backward trajectories originated in the Sahara, spent a few days above it and arrived at CVAO from

a northwest direction. The frequency plot also reveals that backward trajectories featured paths over southern Europe and the Mediterranean region so that anthropogenically emitted particles from these regions might also be carried to CVAO.

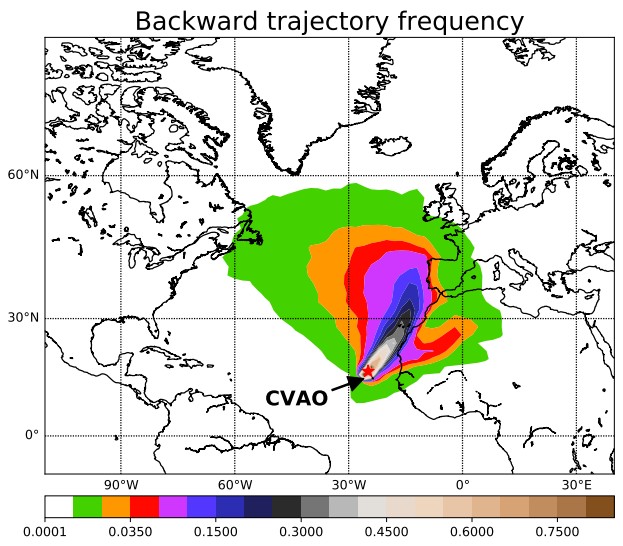

**Figure 6.** Relative frequency of backward trajectories arriving at CVAO.

## 3.2 Particle classification, origins and its monthly variation

### 3.2.1 Particle classification, origins

From the overview over the data sets introduced above, it can be seen that PNSDs, PNC, $N_{CCN}$ and air mass origin show large variability. In the following, we examine how different air mass origins and histories are related to different aerosols being present at CVAO. Therefore, we implement a K-means algorithm to classify aerosol types according to the shape of PNSDs. The input parameter $x_i$ for K-means was defined as the normalized PNC in different size ranges. We used PNC in the size ranges from 20 to 40 nm ($N_{20\_40nm}$), 40 to 80 nm ($N_{40\_80nm}$), 80 to 600 nm ($N_{80\_600nm}$), 600 to 1000 nm ($N_{600\_1000nm}$) and larger than 1000 nm ($N_{>1000nm}$). As the absolute numbers in different modes are very different, we normalized PNC in different size ranges to make sure that all size ranges are assessed equally. The normalization method can be found in Appendix C.

These size ranges were chosen based on assuming that particles in different size ranges feature different formation routes and behaviors. For the here presented data, higher $N_{20\_40nm}$ values indicated the presence of relatively freshly formed particles. The boundary between Aitken and accumulation is roughly at 80 nm at CVAO. Therefore, $N_{40\_80nm}$ and $N_{80\_600nm}$ represented Aitken and accumulation mode particle number concentration, respectively. $N_{600\_1000nm}$ can be attributed to sea-salt with a possible contribution from dust particles (Schladitz et al., 2011). The largest observed particles ($N_{>1000nm}$) are directly linked to heavy dust plumes at CVAO with a small contribution from large sea spray aerosols (Gong et al., 2020b). A summary of input parameters and the corresponding aerosol mode and source is shown in Tab. 2.

**Table 2.** The input parameter $x_i$ for K-means and the corresponding main aerosol mode and source

| Parameter | Main aerosol mode |
|---|---|
| $N_{20\_40nm}$ | Freshly-formed particles |
| $N_{40\_80nm}$ | Aitken mode particles |
| $N_{80\_600nm}$ | Accumulation mode particles |
| $N_{600\_1000nm}$ | Mainly sea spray aerosol with possible dust fraction |
| $N_{>1000nm}$ | Mainly dust, but also large sea spray aerosol |

As mentioned above and described in detail in Appendix C, in this study we determined $k$=5 for the number of clusters. The resulting classification of PNSDs and particle volume size distributions (PVSDs) are shown in Fig. 7. The solid lines show the median of PNSDs and PVSDs of the 5 different aerosol clusters, with a linear scaling on the y-axis in Fig. 7 (a), (c) and a logarithmic in Fig. 7 (b), (d). The error bars indicate the range between 75th and 25th percentiles. Additionally, Figure 8 shows the LAC data and Figure 9 shows the backward trajectory path frequency and backward trajectory height frequency which are connected to these 5 different clusters of PNSDs.

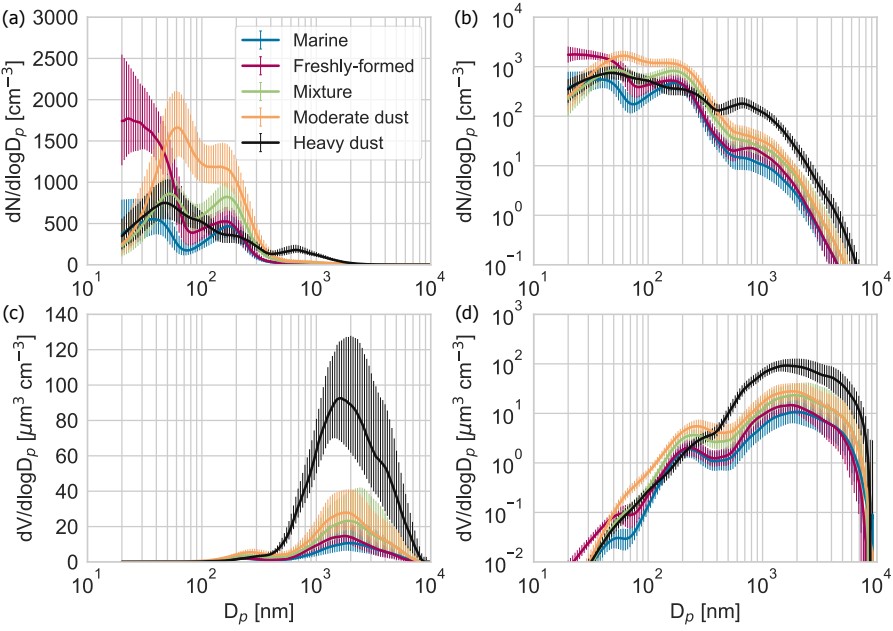

**Figure 7.** The median of PNSDs (a and b panel) and PVSDs (c and d panel) of marine (blue), freshly-form (purple), mixture (green), moderate dust (brown) and heavy dust (black) clusters with a linear (a and c panel) and a logarithmic (b and d panel) scaling on the y axis. Error bars show the range between 75th and 25th percentiles.

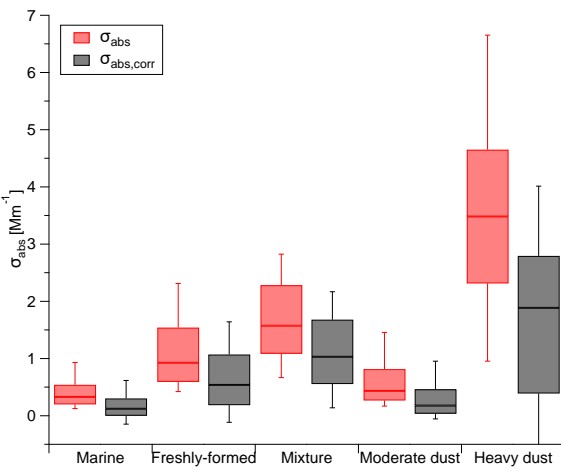

**Figure 8.** The median and 25th and 75th percentiles of LAC data which were sorted according to the classification obtained from the PNSDs.

The marine type PNSD features three modes, i.e., Aitken, accumulation and sea spray aerosol, shown as the blue line in Fig. 7. The marine type featured the lowest $N_{40\_80nm}$, $N_{80\_600nm}$, $N_{600\_1000nm}$ and $N_{>1000nm}$ values. The Hoppel minimum of marine type PNSDs is at roughly 70 nm. During the marine period, the sea spray aerosol mode can be assumed to be primary marine aerosols, as discussed in Modini et al. (2015) and Gong et al. (2020b). The PNSD representing the freshly-formed

cluster is shown as a purple line in Fig. 7. This cluster featured the highest $N_{20\_40nm}$. While we did not measure PNSDs below 20 nm, it can still be safely assumed that such high concentrations of small particles indicate new particle formation events which must have happened recently in the respective air masses, which is known to occur e.g., in the marine free troposphere, followed by downward mixing of the particles (Korhonen et al., 2008; Merikanto et al., 2009). It is interesting to note that this is related to a generally low aerosol particle volume concentration (see Fig. 7 (c) and (d)), as well as mass concentration, as only

comparably small numbers of larger particles are present. This is similar to the above described marine type. So new particle formation seems to happen predominantly in marine air masses with low particle volume concentrations, an observation which has also been made for Barbados in Wex et al. (2016). Furthermore, times during which marine type PNSDs appeared are characterized by the lowest measured values for $\sigma_{abs}$ and $\sigma_{abs,corr}$, in line with the absence of absorbing particles as e.g., black carbon or dust.

The frequency of backward trajectory paths that were observed during times when PNSDs from marine and freshly-formed clusters prevailed, present a similar pattern, as shown in Fig. 9 (a1) and (b1). The majority of these backward trajectories spend the whole 6-day period over the North or western North Atlantic Ocean and arrived at CVAO from the northeast direction. Some of these backward trajectories touched southwest Europe and crossed the coastal areas in northwest Africa.

No clearly visible Hoppel minimum is present for the moderate dust cluster, as shown in a brown line in Fig. 7 (a) and

(b). The moderate dust cluster featured the second highest $N_{coarse}$. We also observed a very high concentration of Aitken and accumulation mode particles for the moderate dust cluster. The heavy dust cluster is shown as the black line in Fig. 7.

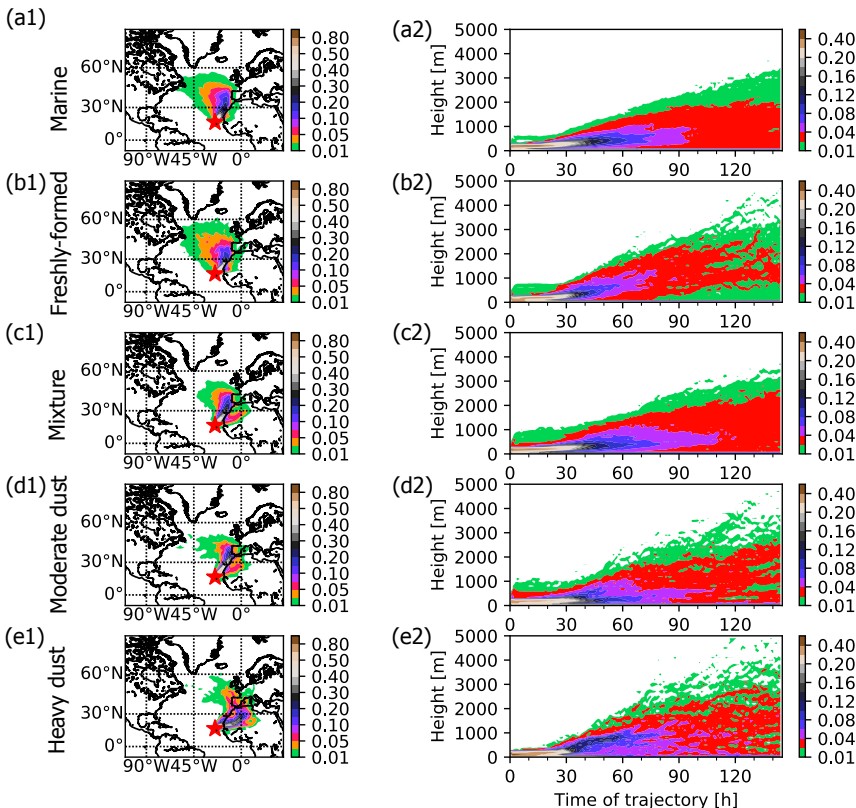

**Figure 9.** Relative frequency of backward trajectory paths arriving at CVAO during marine (a1), freshly-formed (b1), mixture (c1), moderate (d1) and heavy dust (e1) periods and the corresponding relative frequency of backward trajectory height in panel (a2) to (e2).

No clearly visible Hoppel minimum is present for this cluster, either. The heavy dust cluster featured the highest $N_{\text{coarse}}$ and also the highest values of $\sigma_{abs}$ and $\sigma_{abs,corr}$. It is worth to be mentioned that similar PNSDs were found in previous field measurements at the Sahara (Weinzierl et al., 2009; Kandler et al., 2009; Kaaden et al., 2009). The highest $N_{\text{coarse}}$ also means that the highest volume and mass concentrations (Fig. 7 (c) and (d) black line) are present in the heavy dust cluster, which was already reported in a previous study (Fomba et al., 2014).

Finally, the mixture cluster PNSD is shown as green lines in Fig. 7, with three clearly distinguished Aitken, accumulation and coarse modes. $N_{>1000nm}$ are between the marine and dust clusters. The Hoppel minimum of the mixture cluster is at ∼90 nm. This may indicate that for this cluster the maximum supersaturation reached in the trade wind clouds is lower than the marine and freshly-formed clusters, for which, as said above, it was at 70 nm. These two clusters have lower number concentrations for larger particles. Therefore it does not surprise that PVSDs of the mixture cluster are between the marine and dust clusters. In the presence of higher particle concentrations for larger particles, an overall higher amount of water condenses onto particles at cloud base, which can be expected to lead to an overall lower supersaturation (Reutter et al., 2009), enabling only larger particles to be activated to cloud droplets, as observed in our data sets.

The frequency of backward trajectories during the moderate dust cluster featured a similar pattern as during the mixture period (as shown in Fig. 9 (c1) and (d1)). There are two major paths during mixture and dust periods. One class of backward trajectory was from the North Atlantic Ocean and crossed the coastal areas in northwest Africa. One class of backward trajectories spend some days over the Sahara. So overall, for PNSDs which were characterized to belong to the mixture or moderate dust clusters, also sea spray particles contributed overall, and likely also to the particles at larger sizes. The backward trajectories during the heavy dust period were mainly from the Sahara, which justifies assuming that the related particles in this highest $N_{\mathrm{coarse}}$ came from Saharan dust particles.

The relative frequency of backward trajectory heights is shown in the right panels of Fig. 9. Gong et al. (2020b) measured that marine boundary height at CVAO varied roughly from 550 to 1100 m. From Fig. 9 it can be seen that a large fraction of air masses came from or had contact with the free troposphere. In this context, it is worth noticing that it has been described in the past, that new particle formation in the marine environment often takes place in the free troposphere, followed by growth of these particles into the CCN size range (Korhonen et al., 2008; Merikanto et al., 2009).

While, as mentioned above, $\sigma_{abs}$ and $\sigma_{abs,corr}$ were the lowest for the marine type cluster and the highest for the heavy dust cluster, the pattern showing up for these LAC data (Fig. 8) are not fully conclusive. It is striking that the moderate dust cluster has the second-lowest LAC values, although one could assume that the moderate and heavy dust clusters would be more similar in this respect. On the other hand, the freshly formed cluster, which resembles the marine cluster plus additional small freshly formed particles, has much higher LAC values than the marine cluster, while strong absorption by freshly formed particles is not to be expected. We assume that large uncertainties presented by the influence of light scattering are causing these issues. Therefore, this suggests that further measurements with methods that are not influenced by the light scattering coefficient are necessary. These can be photo-acoustic photometers for measuring the light absorption coefficient or Single Particle Soot Photometers (SP2) for measuring the refractory black carbon.

### 3.2.2 Monthly variation

In order to understand the large temporal variability of aerosol types on a monthly basis, the monthly variation of the relative frequency of occurrence, number fraction and volume fraction of the five different aerosol clusters are shown in Fig. 10 (a), (b) and (c), respectively. It can clearly be seen that relative frequency of occurrence, number fraction and volume fraction feature different behaviors. For example, the heavy dust cluster represented 14.6% of the cases encountered in January, contributed only 11.7% to particle number, however about 40.4% to particle volume/mass. This needs to be considered when relating number-based properties such as CCN and INP concentrations to e.g. the mass-based chemical composition.

The heavy dust cluster is mainly present during wintertime. This is because, during wintertime, dust is transported westward from the Sahara in the lower troposphere. The deposition of dust particles takes place over the eastern tropical Atlantic and Cabo Verde region. During springtime, the marine aerosol is more often observed. This is because, during springtime, the CVAO mostly receives North Atlantic marine air parcels along with the NNE trade winds.

It is also worth noticing that the freshly formed cluster has its largest contributions by number in the time from November until April. The reasons for this, however, remain unresolved, as additional measurements of long-term meteorological or gaseous precursors at CVAO are not available.

The above discussed marine, mixture and moderate dust aerosol clusters were recognized from Gong et al. (2020b), who characterized the sources of the particles at CVAO during September and October 2017. The freshly-formed and heavy dust clusters did not show up during the previous study and are described here for the first time for CVAO. It is worth mentioning that even in the relative simple dust-marine intersect location at Cabo Verde, five different aerosol clusters were classified, and these 5 clusters featured different characteristics such as size distributions and origins.

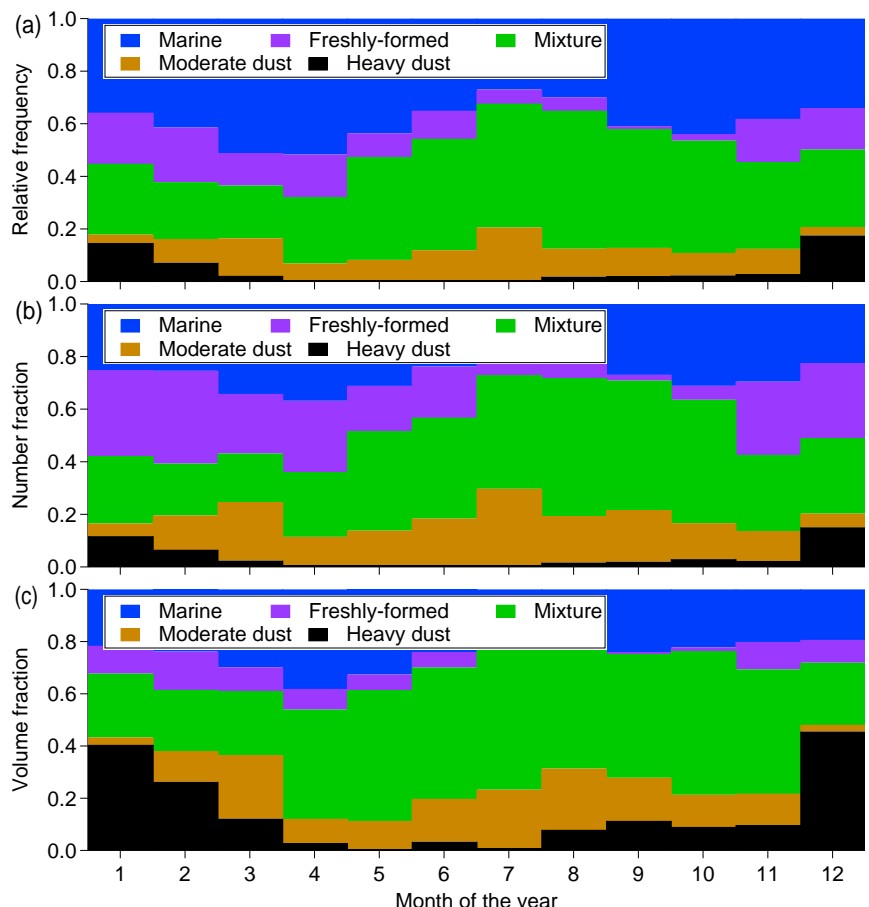

**Figure 10.** Monthly variation of relative frequency (a), number fraction (b) and volume fraction (c) of different aerosol clusters.

### 3.3 Evaluate marine and dust contribution to CCN

One important purpose of this study is to evaluate CCN population and particle hygroscopicity for marine or dust aerosol dominated environments. Therefore, we compared $\kappa$ and $N_{\mathrm{CCN}}$ during the marine and the dust periods, where for the latter, moderate and heavy dust were combined. Figure 11 (a) shows a boxplot of $N_{\mathrm{CCN}}$ at different supersaturations during 9 months with CCNC measurement. $N_{\mathrm{CCN}}$ at a supersaturation of 0.30% (as a proxy for the average maximum supersaturation encountered in clouds) during the dust period is about 2.5 times higher than that during marine periods. $N_{\mathrm{HM}}$ can be used to estimate atmospheric $N_{\mathrm{CCN}}$ at a supersaturation of 0.30%. Therefore, to extend atmospheric $N_{\mathrm{CCN}}$ from 9 months to 10 years, we also derived $N_{\mathrm{HM}}$ for marine and dust periods from 2008 to 2018, and also show these values in Fig. 11 (b). $N_{\mathrm{CCN}}$ at a supersaturation of 0.30% in Fig. 11 (a), derived from only 9 months of measurements, and $N_{\mathrm{HM}}$ in Fig. 11 (b) derived for 10 years agree well, giving confidence in the extension of the CCN data set. Also, it is clear that the atmospheric $N_{\mathrm{HM}}$, which best represents number concentrations to be expected for CCN in the atmosphere, is about 2.5 times higher during dust periods than during marine periods at CVAO.

Besides, $N_{\mathrm{CCN}}$ during dust periods is generally higher than that during marine periods, increasingly so with increasing supersaturation or decreasing $d_{\mathrm{crit}}$. For example, at a supersaturation of 0.70%, corresponding to a $d_{\mathrm{crit}}$ of about 50 nm, $N_{\mathrm{CCN}}$ during dust periods is four times higher than during marine periods. This is interesting, as in the size range around 50 nm dust particles are not expected to be present in significant numbers. An enhanced Aitken mode particle number concentration was observed during moderate dust periods, showing up clearly separated from one large mode observed for particles > 100 nm. We interpret these particles to stem from new particle formation in air masses associated with the dust, a phenomenon which has been observed in a previous field study (Dupart et al., 2012). However, this is highly speculative and proving or disproving this hypothesis is beyond the scope of this paper.

Figure 11 (c) shows $\kappa$ as a function of $d_{\mathrm{crit}}$. The error bars of $d_{\mathrm{crit}}$ show one standard deviation and error bars of $\kappa$ show one geometric standard deviation. $\kappa$ during the marine period generally agreed with that of the dust period within uncertainty when considering the error bars. Therefore, the CCN-derived hygroscopicity for dust and marine periods in the size range between 40 and 140 nm shows no significant difference. This is a surprising but also important result, as this may suggest the use of generalized values for $\kappa$ for related cases. A somewhat comparable result of similar $\kappa$ for differing aerosol was obtained recently by Jayachandran et al. (2022), examining aerosol during the Indian summer monsoon: Their $\kappa$ values scattered over a wide range. But when comparing mean and median values for dry and wet periods they were similar, although total particle number concentrations and $N_{\mathrm{CCN}}$ varied strongly.

Getting back to our results, as stated above, our $\kappa$ values are all representative of the months from October until March. They are comparably small and in agreement with the modeling study by Pringle et al. (2010). The majority of particles in marine air masses may originate from new particle formation, not from sea spray (Korhonen et al., 2008; Merikanto et al., 2009), explaining the derived $\kappa$ which is lower than for sea salt particles. Dust particles, on the other hand, were reported to show hygroscopic behavior in the past (Karydis et al., 2011) and may also get more hygroscopic during aging. However, for dust type aerosols the nature of particles in the size range for which $\kappa$ was determined, particularly for the size range < 100 nm,

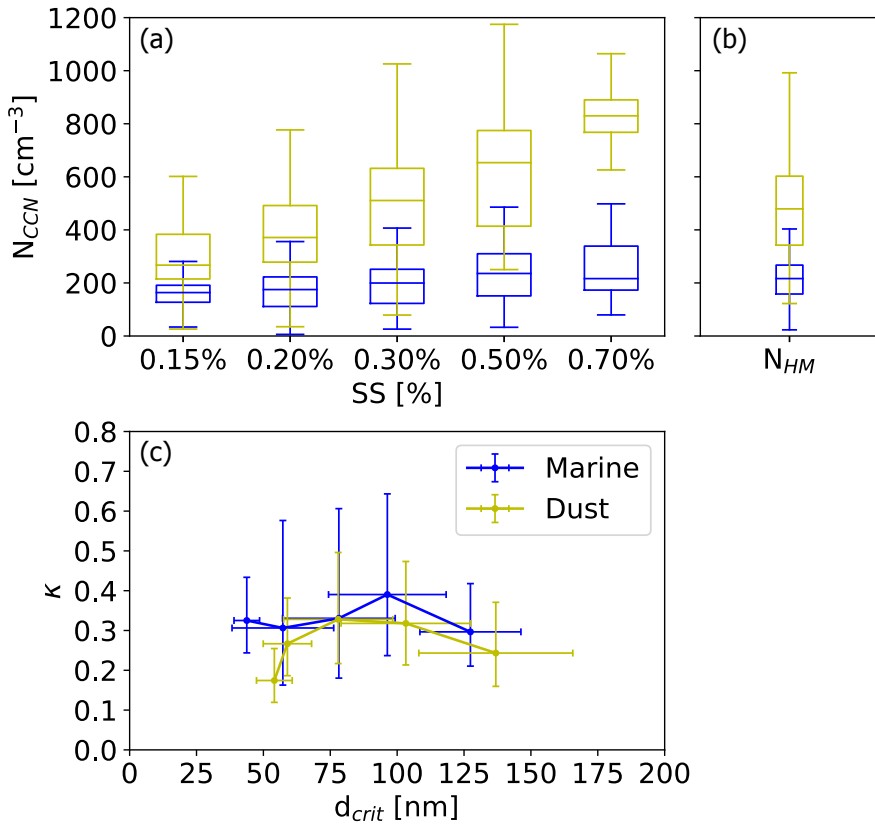

**Figure 11.** (a) Boxplot of measured $N_{CCN}$ during marine and dust periods at supersaturations (SS) from 0.15% to 0.70%; (b) Boxplot of $N_{HM}$ during marine and dust periods; (c) $\kappa$ as a function of $d_{crit}$ during dust and marine periods. Dust and marine periods are shown in yellow and blue, respectively.

is not fully clear, as mentioned above. But it did become clear here that the observed particles do not differ in hygroscopicity from those observed in marine type aerosols, which strengthens our suggestion that these particles also originate from new particle formation.

## 4    Summary and conclusions

5    Long-term data sets as the one presented in this study can yield valuable insights for our general understanding but also for improving the performance of models (Reddington et al., 2017). In this study, we summarized measurements of the aerosol microphysical properties in a representative marine-dust aerosol intersect environment at Cape Verde Atmospheric Observatory (CVAO). In-situ aerosol measurements included light absorbing carbon (LAC), particle number size distribution (PNSD), cloud condensation nuclei (CCN) and corresponding backward trajectories. An unsupervised machine learning algorithm, K-means,

10    was implemented to classify aerosol clusters, which were then the base for the classification of particle sources according to

the shape of particle number size distributions. Based on the classification, we studied the air mass origins for different aerosol clusters and parsed out the marine and dust contributions to CCN and their hygroscopicity.

After classification based on machine learning, we found five different aerosol clusters for the 10 year-long data set obtained at CVAO – marine, freshly-formed, mixture, moderate dust and heavy dust. When examining monthly variations, we showed that heavy dust was mainly observed during wintertime, where it was present roughly 10% of the time, and contributed, on a monthly base, roughly 10% of particle number but up to $> 40\%$ of the particle volume and mass. Moderate dust events, however, occurred throughout the year, typically for more than 10% of the time during each month, contributing up to more than 20% of all particles in number. Fall and winter months were those during which also the aerosol absorption was highest, and in the absence of local anthropogenic pollution sources upwind of CVAO, the observed absorption can be assumed to originate from dust particles.

A special feature of the moderate dust type aerosols was a pronounced Aitken mode, which, as said above occurred frequently. While the determination of the nature of these particles is beyond the scope of this study, we speculate if new particle formation events may also occur in moderately dust laden air masses, based on not yet known processes, and propose this topic for further research.

Clean marine periods were most frequently present during springtime, occurring roughly 50% of the time in March and April. Periods during which the freshly-formed type aerosol prevailed had similar air mass origin as marine aerosol, indicating that the typical and well-known way of new particle formation mainly occurs in marine aerosols with low particle volume concentration and low particulate surface area. Together, clean marine and freshly-formed air masses occurred for more than 50% of the time from October until May, while the mixed type aerosol was the most often occurring type in the remaining months.

Dust and marine type aerosols featured clearly different PNSDs. Enhanced particle number concentrations and CCN number concentrations were observed when air masses came from the Sahara. For air masses originating from the Atlantic Ocean, PNSDs featured the lowest number concentrations in Aitken, accumulation and sea spray mode. CCN number concentration at a supersaturation of 0.30% (a proxy for the average maximum supersaturation encountered in clouds) during the marine period was about 2.5 times lower than during the dust period. However, the CCN-derived particle hygroscopicity (in the size range from 40 to 140 nm) for marine and dust periods showed no significant difference, independent of the air mass. This latter result, however, was based solely on data from the months of October until March. Averaging all $\kappa$ values obtained in this study, an average value of 0.32 was found, again for the months from October until March, and omitting small size-dependent differences which, however, all were within uncertainty.

The Hoppel minimum, a dip between the Aitken and accumulation mode indicating the division between particles that had been activated to cloud droplets from those that had not, was found to be at roughly 80 nm. More precisely, the Hoppel minimum was at 70 nm for marine and freshly-formed type aerosols and at 90 nm for the mixture type aerosol, where the latter had higher number concentrations for larger particles. This larger size for the Hoppel minimum likely is indicative of these particles being a larger sink for water vapor during cloud droplet formation, lowering the effective supersaturation for air masses with mixture type aerosol.

## 5 Outlook

We proved that an unsupervised machine learning algorithm can be a good tool to classify aerosol types. It benefits from the long-term measurements with extremely large data sets. Moreover, we point out that the summarized data sets can also help to shape new research questions towards future studies at CVAO. The freshly-formed particles and the corresponding backward

trajectories indicated one type of new particle formation which happened at CVAO mainly in air masses of marine origin. The height of backward trajectories indicated that new particle formation might have occurred in the free troposphere and particles were then transported down to the marine boundary layer. While new particle formation in the marine environment was assumed to occur in the free troposphere before, we still lack the knowledge of the overall mechanism, such as the precursors, growth rate, temperature and humidity effects of new particle formation, and also the fraction of CCN which originate from new

particle formation. Further, the moderate dust period featured high concentrations of Aitken mode particles, while the heavy dust period did not. More investigations are required to understand the reason behind that.

The data sets collected at CVAO are invaluable as, due to their long duration, they can help to better understand the aerosol microphysical properties, which are at the base of aerosol-cloud interactions. Furthermore, the data sets represent highly valuable information for driving, evaluating and constraining atmospheric model simulations. Long-term measurements of aerosol

size, light absorbing carbon and CCN are important in the future because the real world has a large spatial heterogeneity and temporal variability. Additionally, with our data being publicly available, it can be included for more in-depth studies focussing on differences such as between seasons or similar months in different years or other more distinct topics. A wealth of data is already available in general and will hopefully be exploited in more detail in the future.

*Data availability.* All data described and analyzed in this study are published in the Data Publisher for Earth & Environmental Science

(PANGAEA) (https://doi.org/10.1594/PANGAEA.921321, Gong et al., 2020).

## Appendix A:  Particle loss

We used the Particle Loss Calculator (written in IGOR Pro, von der Weiden et al., 2009) to calculate the particle losses within the inlet tube system. The aerosol sampling inlet system includes a vertical stainless-steel sampling pipe (32m long, 1/2 in. outer diameter), a diffusion dryer (directly on top of the measurement container), and $\sim$3 m conductive silicon tubing (3/8

in. inner diameter). We considered size-dependent particle losses due to sedimentation, diffusion, turbulent inertial deposition, inertial deposition in a contraction and in a bend. Note that the losses inside the diffusion dryer in this study are equivalent to the losses in a 15 m long tube (Wiedensohler et al., 2012). Figure A1 shows the resulting particle losses (in %) as a function of particle size.

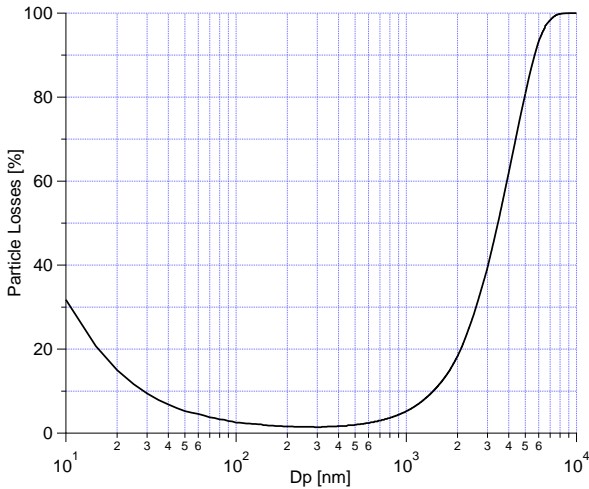

**Figure A1.** Size-dependent particle loss through the inlet at CVAO station.

## Appendix B: MAAP correction

Figure B1 shows the PDF of $\sigma_{abs}$ (red line) and $\sigma_{abs,corr}$ (black line). Although corrections for the scattering artifact for filter-based absorption photometers are highly uncertain, the results clearly underline the importance of the correction.

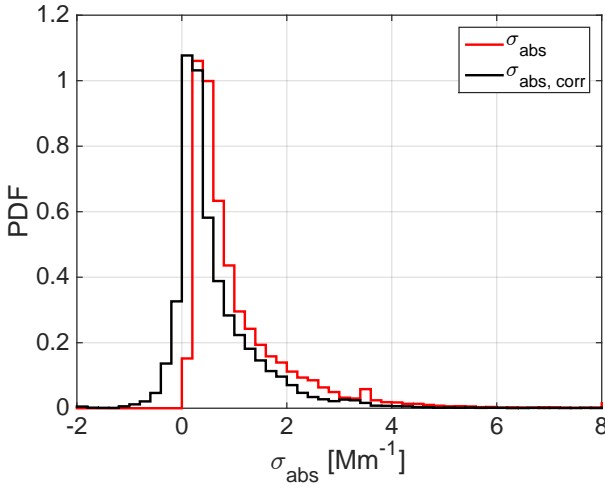

**Figure B1.** Probability density functions (PDFs) of $\sigma_{abs}$ as red line and $\sigma_{abs,corr}$ as black line.

## Appendix C: Input parameters normalization and determine classification clusters

To make sure PNC in all size ranges are assessed equally when applying the K-means algorithm, we normalized PNC in different size ranges separately based on a min-max normalization:

$$x_i = \frac{x_i - x_{min}}{x_{max} - x_{min}} \tag{C1}$$

where $x_i$ is $i_{th}$ PNC value in a specific size range. For PNC in a specific size range, the minimum PNC value of that size range gets transformed into a 0, the maximum PNC value gets transformed into a 1, and every other value gets transformed into a decimal between 0 and 1.

The elbow method is a useful graphical tool to estimate the optimal number of clusters $k$ for a given task. Theoretically, if $k$ increases, the within-cluster sum of squared errors (a.k.a. distortion) will decrease. This is because the samples will be closer
to the centroids they are assigned to. In the elbow method, distortion begins to decrease rapidly and then switches to a more slowly decrease at a certain $k$ value. Figure C1 shows the elbow plot as a blue line. The distortion started decreasing in a linear fashion after $k$=5. Therefore, $k$=5 is the optimal number of clusters. Also, we used the Calinski-Harabasz score (Caliński and Harabasz, 1974), which computes the ratio of dispersion between and within clusters. The vertical black dashed line indicates the highest Calinski-Harabasz value which occurred, which also is at a number of five clusters. Therefore, overall, the optimal
number of clusters was derived to be five, here, which is the value we then used within this study.

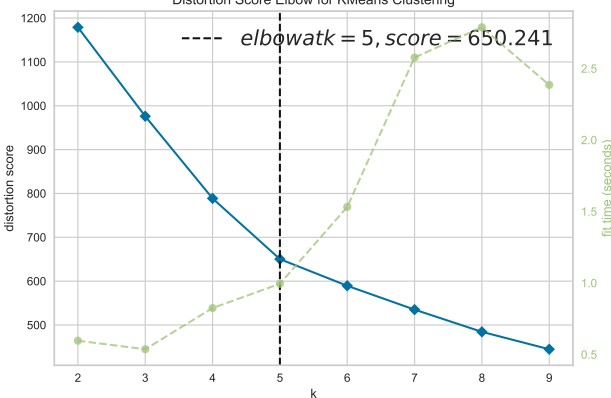

**Figure C1.** The distortion as a function of $k$ (a.k.a. elbow plot) is shown as blue line. The highest Calinski-Harabasz score is shown as dashed black line. The consumed time as a function of $k$ is shown as dashed green line.

## Appendix D: Correlation between $N_{\text{CCN,0.30\%}}$ and $N_{\text{>HM}}$

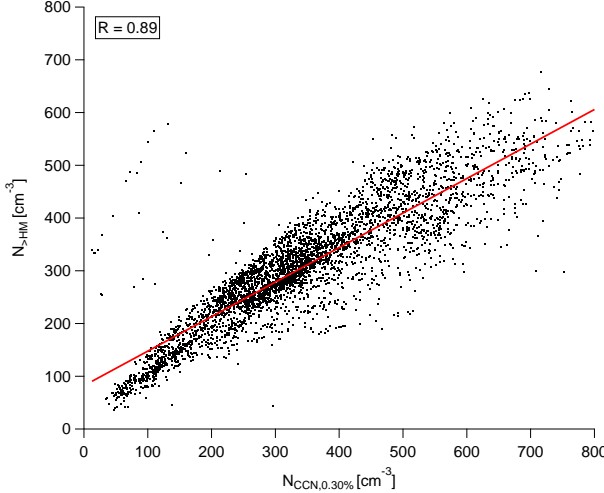

**Figure D1.** Scatter plot of $N_{CCN,0.30\%}$ against $N_{>HM}$ in black dots. The linear fitting line is shown in red line.

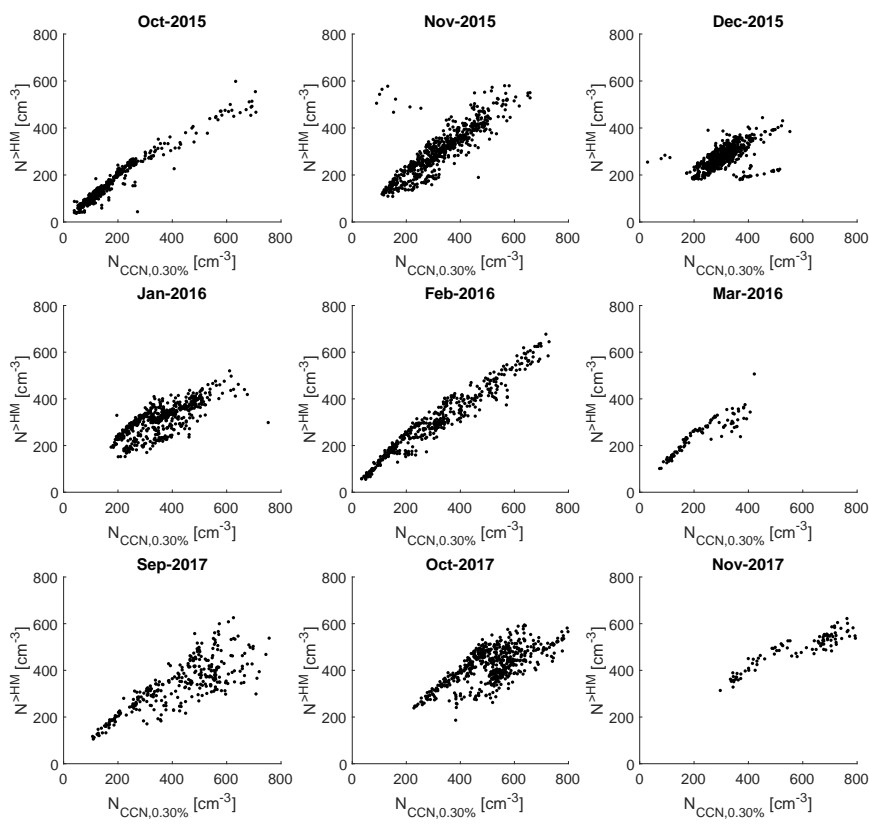

**Figure D2.** Scatter plot of $N_{CCN,0.30\%}$ against $N_{>HM}$ during each month in black dots.

*Author contributions.* XG conceived the study and wrote the manuscript with contributions from HW, TM and FS. TM performed particle number size distribution and light absorbing carbon measurements. SH performed cloud condensation nuclei measurement. XG evaluated and analyzed the data and developed analysis methods. XG, HW, TM, SH, JV, AW and FS discussed the results. All of the co-authors proofread and commented the manuscript.

5 *Competing interests.* The authors declare that they have no conflict of interests.

*Acknowledgements.* The authors acknowledge the Leibniz Association SAW funding for the project "Marine biological production, organic aerosol particles and marine clouds: a Process Chain (MarParCloud)", SAW-2016-TROPOS-2. We acknowledge the SOPRAN (Surface Ocean Processes in the Anthropocene) project, supported by the Bundesministerium für Bildung und Forschung (BMBF) under grant number 03F0611J. We acknowledge the SAMUM (Saharan Mineral Dust Experiment) project, founded by the German Research Foundation under 10 grant MU 2669/1–1 and grant FOR 539. The authors acknowledge the CVAO site manager, Luis Neves, to maintain SMPS, APS and CCN measurements at CVAO. We acknowledge Thomas Conrath and Kay Weinhold for their technique support.

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
