# Peer review of "Understanding aerosol microphysical properties from 10 years of data collected at Cabo Verde based on an unsupervised machine learning classification"

_Atmospheric Chemistry and Physics, 2021_

## Author Comment (AC1)

Dear Reviewer,

We thank you for doing this review and for your suggestions that helped to improve our manuscript. Below, please find your original comments in blue and our responses in black. When referencing page and line numbers, we are always referring to the new version of the manuscript.

The paper 'An unsupervised machine-learning-based classification of aerosol microphysical properties over 10 years at Cabo Verde', by Gong et al. investigates aerosol properties and their relation to properties relevant for cloud formation and puts the results also in the perspective of air mass origin. This topic is very relevant for a better understanding of the interplay between aerosol and clouds. In particular, this study investigates data sets for a long time period and for a region of interest (influenced by both marine and dust sources, area with not so many observations). The manuscript is therefore well fitting into the scope of Atmospheric Chemistry and Physics.

Overall, the manuscript is well written and I can recommend publication for ACP after some revision, described below.

I have a slight preference that sections 4 and 5 are re-organised. The synopsis presents already to a good part the conclusions, and the conclusions more the future work. I would prefer to have a well-structured section 4 as conclusion and a shorter section 5 for the future work outlook.

We merged original sections 4 and 5 as " Section 4 Summary and conclusions". In the new version, section 5 is Outlook.

General comments

As machine learning is in the title, I would have expected a more detailed introduction to it. However, for the understanding of the article, the necessary descriptions are given. But the authors should explain also the difference between supervised and un-supervised machine learning algorithms.

We added the following content towards the end of the introduction (page 3, line13 in the new version of the manuscript) to give more information about supervised and unsupervised machine learning algorithms.

"The machine learning algorithms include supervised and unsupervised approaches, depending on using labeled or unlabeled data to help predict outcomes. In general, supervised machine learning requires upfront human intervention based on prior knowledge on a connection between input data and desired output values. By feeding data and desired outputs into an algorithm, weights are adjusted in the model until the output is met sufficiently well, and can then be used for characterizing further input data. On the other hand, unsupervised machine learning

algorithms learn the inherent structure of the data without using explicitly provided labels. The goal is to infer the natural structure present within a set of data points."

Also, the title may have overemphasized the machine learning algorithm. We changed the title to:

"Understanding aerosol microphysical properties from 10 years of data collected at Cabo Verde based on an unsupervised machine learning classification".

The light absorbing carbon data has not been included in the aerosol type classification. However, such a long data set would be worth to study or use in more detail. Have the authors tested to include the LAC data in the clustering?

The light absorbing carbon (LAC) data originally was not included in the classification as

a) no LAC data was available for 17.52% of the measurement period for which particle number concentration data is available, and b) we wanted to focus on the more often used and more widely needed particle number size distributions.

However, as this topic was mentioned in both reviews, we now included and discussed the LAC data sorted according to the derived clusters in Sect. 3.2.1 and a new figure (Fig. 8 in the new version of the manuscript). The main part of this discussion is at the end of this section and is as follows:

"While, as mentioned above, $\sigma_{abs}$ and $\sigma_{abs,corr}$ were the lowest for the marine type cluster and the highest for the heavy dust cluster, the pattern showing up for these LAC data (Fig. 8) are not fully conclusive. It is striking that the moderate dust cluster has the second-lowest LAC values, although one could assume that the moderate and heavy dust clusters would be more similar in this respect. On the other hand, the freshly formed cluster, which resembles the marine cluster plus additional small freshly formed particles, has much higher LAC values than the marine cluster, while strong absorption by freshly formed particles is not to be expected. We assume that large uncertainties presented by the influence of light scattering are causing these issues. Therefore, this suggests that further measurements with methods that are not influenced by the light scattering coefficient are necessary. These can be photo-acoustic photometers for measuring the light absorption coefficient or Single Particle Soot Photometers (SP2) for measuring the refractory black carbon."

The authors dispose of a 10 years long data set. Although a break-down by year or seasons and respective description would probably result in a too long paper, it would could have been worth to analyse this for some distinct topics. E.g., the interesting result of high Nccn numbers during dust periods with low critical diameter (as discussed on page 17, lines 1-6). Maybe a look per year or by season would have brought some additional insights to this.

Indeed, as the reviewer suggests, a more detailed examination may reveal important additional insights, but, also as the reviewer assumes, this is beyond the scope of this study. It would prolong this already long manuscript even more.

However, as the data are now freely available, maybe this topic will be picked up in the future, and we explicitly suggest this at the end of our text now.

Page 2, line 2: please check also for a few more recent dust INP articles (eg, Hoose et al, doi:10.5194/acp-12-9817-2012.Kanji et al, https://doi.org/10.1175/AMSMONOGRAPHS-D-16-0006.1, Boose et al., https://doi.org/10.5194/acp-19-1059-2019)

Thanks for pointing this out! However, as this work is not on INP, we only added the review by Kanji et al., 2017.

Page 3, line 3: the machine learning algorithms are described to deliver 'faster, accurate results' this is a comparison, please mention compared to what.

We here compared machine-learning algorithms with traditional statistical methods. We included this in the text, together with the additional text we referred to above.

Page 3, lines 1 to 10: the aerosol classification is done by machine learning algorithms. Not mentioned is why other methods, applied often like, eg, multivariate analytical method, principal component analysis were not applied here.

We now introduce the difference between supervised and unsupervised machine learning in the introduction section. As we wanted to explore the inherent structure of the data without influencing the outcome, i.e., without using explicitly-provided labels, unsupervised machine learning algorithms were the better choice, compared to supervised machine learning algorithms.

We do know that machine learning is a "melting pot" of different algorithms because of its massive diversity. Figuring out the difference between machine learning algorithms, and evaluating their performance are not the topics of this study. However, we used half-year data to test the performance between k-means and DBSCAN (Density-Based Spatial Clustering of Applications with Noise), and both methods show comparable results. But the DBSCAN output contained some outliers and noisy datasets, aggravating classification. Most importantly, DBSCAN is not suitable for big datasets (as we have 10-year data) and has high computational complexity.

To the best of our knowledge, principal component analysis (PCA) is an unsupervised, non-parametric statistical technique primarily used for dimensionality reduction in machine learning.

This method could have been used before we performed k-means if we would have had too many input parameters. But generally, this method is not suitable for this study.

Multivariate analysis methods are used in the evaluation and collection of statistical data to clarify and explain relationships between different variables. We learned that multivariate testing is widely used for webpage design, such as helping to target redesign efforts to the elements of a webpage where they will have the most impact. We did not test the possibility of multivariate analysis for this study.

Page 3, lines 13-17: would it be possible to integrate a wind rose graph here? In Gong et al 2020b there is one, however, not for the whole time period investigated here. Please refer to a suitable reference (like done in Gong et al 2020b). In addition, for the air mass origin analysis it is mentioned later that the boundary layer height was derived in previous studies. The paper of Gong et al 2020b referred to, is however only for a short period. The study here is however over 10 years. How can the authors assure that the used assumption for the boundary layer height is overall valid?

Unfortunately, meteorological data for the examined period is not available. But Carpenter et al. (2010) reported the monthly wind rose plot from January 2007 to December 2009 for the Cape Verde Atmospheric Observatory (CVAO) and found that the predominant wind direction was from the northeast (see Fig. 9 in Carpenter et al., 2010).

Carpenter et al. (2010) also analyzed radiosonde data from October 2007-August 2008 and found that MBL height varied from ~300 to 1500 m at Cabo Verde. Therefore, it is reasonable to use 200 m height as a start point to calculate the back-trajectory.

We cited this study and added the above information in the text.

Page 3, lines 18ff: the aerosol inlet is at 32m; it is however not mentioned in this text how long the total inlet tubing to the measurement container is (mentioned however in Gong et al 2020b). However, this is important to judge the sampling set up. The authors should also mention that this long tubing and related particle losses is accounted for in their particle loss corrections (it is I assume).

This paper and Gong et al. (2020b) used the same setup. The particle losses inside the 32 m long tube were included. We added this information in Section Appendix A: Particle losses.

Page 3, line 19: '… to minimize the influence of sea spray aerosol…': Please explain bit more, why in particular the sea spray aerosol should be minimized, or refer, eg, to set ups at GAW stations like Mace Head, Ireland how they set up such measurements, at which height?

Due to the shoreline effect, sea-spray aerosol, produced by bubbles from coastal oceanic breaking waves, is much more abundant at lower heights than over the open ocean. The

measurement station at the CVAO station is designed to measure aerosol representative of the marine boundary layer environment in the North Atlantic Ocean. For that location, this meant that the sampling inlet needed to be placed on a 32 m high tower.

Typically, for such stations close to the shoreline, aerosol inlets are always placed at large heights. For example, the observation tower of the Atmospheric Research Station at Mace Head has a height of 23 m (http://www.macehead.org/index.php?option=com_content&view=article&id=46&Itemid=27) and is located at a less rugged shoreline, compared to the CVAO. At Ragged Point on Barbados, the aerosol inlet is installed on top of a 17 m high mast towering over 30 m high cliffs (Wex et al., 2016).

We feel that our original description will suffice as an explanation: "The purpose of installing an aerosol inlet on such a high tower is to minimize the influence of sea spray aerosol generated in the surf zone at the coastline."

Page 4, line 8: The authors mention that MPSS and APS were calibrated regularly. Please mention briefly how, where.

We added the following at the beginning of Sect. 2.2.:

"The necessary parameters controlling the sizing of the instruments were logged continuously. Obvious problems were solved by a station engineer. Calibration of sensors took place irregularly but at least once a year and the sizing was checked using traceable PSL particles of sizes 200, 1000, and 2000 nm. Sizing errors are typically below 5%. A traceable check of the counting efficiency is not possible at the station. Instead, instruments calibrated at TROPOS (in the WCCAP - World Calibration Center for Aerosol Physics) were sent to the station infrequently for inter-comparison measurements or, if needed, for replacing instruments."

Page 4, lines 25ff: derivation of scattering coefficient; assumption for the refractive index. The scattering coefficient is not presented later in the manuscript. This might be either skipped or the authors describe why this derivation can be useful for their analyses.

The authors have inserted the following sentences on page 5, line 29:

"However, due to the assumption of a refractive index and the assumption of spherical particles, the quality of calculated scattering coefficients are not sufficiently good, e.g., for the use in radiative transfer calculations. Therefore, scattering data are not presented. The calculated scattering coefficients serve as the best estimate for minimizing artifacts in the absorption measurements."

And at the end of the paragraph, one sentence further:

"It can be seen later in the study (Sect. 3.2.1.), that further improvements for the scattering corrections are of importance, especially for high single-scattering albedos since Eq. 2 is merely

a first order correction. A deeper investigation requires instruments not affected by scattering artifacts, e.g. photo-acoustic photometers."

Page 5, section 2.5, backward trajectories: the authors should give some more details on the initialising parameters for the HYSPLIT model. E.g., which data were used for the meteorological fields, which height resolution was applied, what was the spatial grid size resolution?

The HYSPLIT model used the GDAS (Global Data Assimilation System) meteorological dataset.

GDAS is obtained 4 times a day, i.e., at 00, 06, 12, and 18 UTC. Its spatial resolution is 1-degree latitude-longitude (360 by 181). The GDAS data are output on 23 predetermined pressure surfaces. More information can be found on the NOAA website (https://www.ready.noaa.gov/gdas1.php).

The HYSPLIT is an established model to compute air parcel trajectories. It is widely used for atmospheric science. We added to the paper that calculations were based on GDAS data with a 1-degree spatial resolution.

Page 6, section 3.1.1 and related figure 1: the authors mention that the number concentration for the supermicron particles show a high variation (1 to above 100 (here, the unit is missing)) – but in the figure the scale only goes up to 50 cm-3. Further, in the caption in Figure 1 the submicron range is given for 10 to 1000 nm, but in section 2, the MPSS measures only from 20 nm onwards. Also, please mention the time resolution of the PNSD data – hourly, daily averages?

Thank you for your careful inspection.

In Sect. 3.1.1, page 6, line 18, it is said: "a high variation, from 1 to above 100, with a median value of 3.8 $cm^{-3}$". The unit is "$cm^{-3}$", and we added it again after 100.

The value in the contour plot (Fig. 1) means dN/dlogDp. The concentration of supermicron particle ($N_{>1000nm}$) is the area in the size larger than 1000 nm in the particle number size distribution, see below the schematic plot.

[Figure]

The extremely high $N_{>1000nm}$ values only occurred very rarely, as we mentioned that the maximum is 100 cm$^{-3}$ and the mean value is just 3.8 cm$^{-3}$. Therefore, it is proper to use 50 cm$^{-3}$ as the maximum value in the contour plot. The values larger than 50 cm$^{-3}$ used the same color as 50 cm$^{-3}$. This is a common way to plot such a contour plot.

The lower size cut is 20 nm. We changed the caption of Fig. 1.

The PNSD data is based on the hourly average. We mentioned it in the caption of Figure 1. Now we additionally made a new table (Table 1) to summarize all data information, and included the time resolutions of all datasets.

Page 6, section 3.1.2: line 20: replace 'concentrations' with values. It's the absorption coefficient, not eBC concentration; Please adapt also accordingly in the caption of Figure 2.

Done.

Page 6, section 3.1.3, CCNC time series: please mention on which time basis the values are presented – hourly, daily? Please mention this also in the caption of Figure 3. Further, CCN values for a SS=0.7% are missing for around December 2015 to March 2016 – why?Also, please mention in the caption of Figures 4 and 5 if the shown values are for the whole CCN measurement period (I assume to be so).

CCN number concentrations at each supersaturation are based on hourly data. We added this information in the new Table 1 (which summarizes all measurements), in the caption of Fig. 3, and in the measurement and method description section.

From December 2015 to March 2016, SS=0.70% was not measured. The set-up was sure that each supersaturation scan takes 1 h. The supersaturation was switched every 12 minutes when we measured at five supersaturation measurements, whereas every 15 minutes when only four supersaturations were measured. We updated this information in the Sect. 2.4.

We also added the additional information to the captions of Fig. 4 and 5: "… based on all available data"

Page 9, line 3: Barbados: mention briefly where it is located

Barbados is the easternmost island in the Caribbean. We mention this now in the text.

Page 9, line 9: Hoppel minimum and related supersaturation 0.3%: this is valid for the presented data set, please write accordingly

We added, "at the CVAO during our measurement periods".

Page 9, line 13: comparing Nccn at 0.3% ss with Nhm : the mentioned scatter plot: either insert a 'not shown' here or show it (maybe in appendix); also, please clarify if the correlation is valid for the whole CCN data set period, or how it changes if you look at the correlations per month.

We now included this scatter plot in Appendix D. The correlation coefficient of 0.89 is for the whole dataset.

Not surprisingly, the high correlation between $N_{CCN,0.20\%}$ and $N_{>HM}$ is also observed for every monthly dataset, as can be seen in the following figure. We also included this figure in Appendix D.

[Figure]

Page 11, lines 9-10: how would the number of derived aerosol types differ if the authors would have chosen a different set/number of size range bins? Have the authors tested this? The chosen 5 size ranges take by themselves already a hypothesis of aerosol type classification. Does this not pre-define the result of the classification?

Thanks for this good question.

As summarized in Table 2 (formerly Table 1), particles in different size bins have different formation routes and features. We inferred that based on our background knowledge of different aerosol modes and the respective particles sources. Randomly selecting different size bins would not be useful for data interpretation.

It would also not be helpful for machine learning algorithms when using more size bins. The performance of machine learning algorithms can degrade with too many input variables, especially for the k-means algorithm. K-means algorithm has a difficult time accurately clustering data of high dimensionality (i.e., too many features) because it uses a distance formula to determine cluster membership. When our clustering algorithm has too many dimensions, pairs of points will begin to have very similar distances, thus it can not be able to obtain meaningful clusters.

Page 12, Figure 6: please clarify briefly if the given relative frequency, integrated over the whole, totals finally 1 (then it would not be [%]) or 100 %  / same for Figure 8.

Thanks for your remark. We removed the [%] in the figure. It will add up to 1.

Page 13, line 6: the authors argue that the high concentrations of very small particles indicate new particle formation events. It sounds like as if only NPF events are responsible for these concentration numbers. Could also other mechanisms like, eg, simple transport (from upper troposphere), could be responsible?

Yes, NPF events are frequently observed in the free troposphere, and newly formed particles are entrained from there. NPF events can also happen in the de-coupled marine boundary layer. In general, when we observed the high concentrations of Aitken mode particles on the ground station, these particles already went through condensational growth.

Other sources for high concentrations of small particles elude us. The "simple transport" from the upper troposphere you mention would mean that small particles would need to get there somehow, and for that, NPF is the only source we know.

We amended one sentence here so that it now reads as:

"While we did not measure PNSDs below 20 nm, it can still be safely assumed that such high concentrations of small particles indicate new particle formation events which must have happened recently in the respective air masses, which is known to occur e.g., in the marine free

troposphere, followed by downward mixing of the particles (Korhonen et al., 2008, Merikanto et al., 2009)."

Page 15, section 3.2.2: the authors mention only briefly the freshly-formed cluster in this section. However, from the seasonal cycle in Figure 9 there are clear variations worth to be discussed.

It is very hard to extend the discussion about the freshly-formed cluster because we are missing some important parameters. For example, we did not measure the gas concentrations of, e.g., DMS, SO2, NOx, and VOC. Meteorological data is also important for understanding the mechanisms of NPF but is not available. This is why we mentioned the freshly-formed cluster in the final section of our work in the first place. Now, we added here:

"It is also worth noticing that the freshly formed cluster has its largest contributions, by number in the time from November until April. The reasons for this, however, remain unresolved, as additional measurements of long-term meteorological or gaseous precursors at CVAO are not available."

Page 15, lines 26-30 would also fit into the introduction, in order to tell the reader why this paper makes a significant additional contribution, compared to the previous Gong et al papers for Cabo Verde.

We added the following in the introduction, page 2, line 32.

"Gong et al. (2020b) characterized different aerosol types (marine, mixture, and moderate dust) at CVAO, for data collected during September and October 2017."

And as the second sentence in the next paragraph:

"With this, we extend the characterization by Gong et al. (2020b) by including additional information on seasonal variations and provide additional data on the CCN population and particle hygroscopicity for marine and dust aerosol dominated environments."

Page 18, line 6-8: please clarify in text that the observed particles during dust periods with comparable hygroscopicity like particles during marine periods – that thes were most likely not dust particles, but that the new particle formation happened within the dusty air mass origin events.

We think we already make this very clear in the text, as said "which strengthens our suggestion that these particles also originate from new particle formation" on page 21 line 2.

Page 19, lines 1-7: please clarify if the kappa values discussed here for months October through March are for all the 10 years of PNSD observations (and accordingly with derived Nccn numbers as described with the method earlier).

It is very clear that we are using the kappa values from October 2015 to March 2016 and from October to November 2017, as shown in Figure 3.

Technical comments

Title:  machine-learning:  with or without hyphen? Because, otherwise in the manuscript it is written without hyphen

We removed the hyphen in the title. We also keep it consistent throughout this manuscript.

Page 2, lines 1 to 17: Please check time applied

We checked the paragraphs in question but could not see any misuse of the tense. Please provide us with more concrete clues, or either we will have to assume that the language check in the end will correct possible errors.

Page 3, line 31: '.. for a more detailed explanation…' or '… for more detailed explanations …'

Done.

Page 4, line2: skip 'a' before 'density'  /  line 3: '… chloride are …' / line 4: '… of mineral dust are … and within a range of …' / line 5: '… shape factor and density of 1.17 and 2000 kg m-3 were …'

Done.

Page 4, line 20: 'extent'

Done.

Page 6, line 13: '… particle number concentration (…) in number per cubic…'

Done.

Page 13, line 13: '… were present a similar …' ; discard the 'were'

Done.

Page 17, line 4: '… from new particle formation in an air mass …' or '… in air masses ..' / 'a phenomenon'

Done.

Page 19, line 22: 'K-means' can be skipped here

Done.

**References**

Carpenter, L. J., Fleming, Z. L., Read, K. A., Lee, J. D., Moller, S. J., Hopkins, J. R., Purvis, R. M., Lewis, A. C., Müller, K., Heinold, B., Herrmann, H., Fomba, K. W., van Pinxteren, D., Müller, C., Tegen, I., Wiedensohler, A., Müller, T., Niedermeier, N., Achterberg, E. P., Patey, M. D., Kozlova, E. A., Heimann, M., Heard, D. E., Plane, J. M. C., Mahajan, A., Oetjen, H., Ingham, T., Stone, D., Whalley, L. K., Evans, M. J., Pilling, M. J., Leigh, R. J., Monks, P. S., Karunaharan, A., Vaughan, S., Arnold, S. R., Tschritter, J., Pöhler, D., Frieß, U., Holla, R., Mendes, L. M., Lopez, H., Faria, B., Manning, A. J., and Wallace, D. W. R.: Seasonal characteristics of tropical marine boundary layer air measured at the Cape Verde Atmospheric Observatory, Journal of Atmospheric Chemistry, 67, 87–140, https://doi.org/10.1007/s10874-011-9206-1, http://dx.doi.org/10.1007/s10874-011-9206-1, 2010.

Wex, H., K. Dieckmann, G. C. Roberts, T. Conrath, M. A. Izaguirre, S. Hartmann, P. Herenz, M. Schäfer, F. Ditas, T. Schmeissner, S. Henning, B. Wehner, H. Siebert, and F. Stratmann (2016), Aerosol arriving on the Caribbean island of Barbados: Physical properties and origin, Atmos. Chem. Phys., 16, 14107–14130, doi:10.5194/acp-16-14107-2016.

---

## Author Comment (AC2)

Dear Reviewer,

We thank you for doing this review and for your thoughtful and strict suggestions that helped to improve our manuscript. Below, please find your original comments in blue and our responses in black. When referencing page and line numbers, we are always referring to the new version of the manuscript.

The manuscript investigates the microphysical properties of aerosols based on a large data set at a remote site. They have used an unsupervised algorithm to classify the properties and investigated further, based on the corresponding air mass history also. The study presents a valuable data set for a long duration and follows a novel technique. However, the organization/focus of the manuscript is confusing with respect to the title, along with some other concerns. The results and discussion section could have been made more comprehensive and precise, say, the LAC results are disturbing the continuity between the particle NSD and CCN discussion unnecessarily. Sections 4 and 5 could have merged to form the summary and conclusions. One major concern is regarding the estimation of the effective hygroscopicity parameter and their further interpretations. The paper is worth publishing in the journal of Atmospheric Chemistry and Physics after considering the following aspects.

We merged original sections 4 and 5 as " Section 4 Summary and conclusions". In the new version, section 5 is Outlook.

We extended the discussion of LAC results, more details in the following responses.

The estimation of the effective hygroscopicity parameter and their further interpretations are discussed in the following response.

**General Comments**

- The effective hygroscopicity parameter (Petters and Kreidenweis, 2007) represent the hygroscopicity for that dry diameter. When the critical diameter (obtained from the back-integration of the NSD) is considered as the dry diameter, the corresponding K value should indicate the minimum hygroscopicity of the aerosol system at that supersaturation, since all the particles above that critical diameter should activate as CCN at lower SS itself. So how is the claim in Line 1 on Page 8 valid?

The sentence in question is "The derived $\kappa$ at a given supersaturation represents the particle averaged hygroscopicity around the corresponding $d_{crit}$ ."

As said, $\kappa$ is representative of particles with roughly the size of $d_{crit}$ . It is correct that it does not give any information about particles of clearly differing sizes. But we also do not make this claim.

Also, as stated in the first version of the manuscript already, in the paragraph below Eq. 4, "In this method, aerosol particles are assumed to be internally-mixed". This is a generally used assumption when deriving $\kappa$ like this.

Therefore, within the described restrictions, $\kappa$ does represent the averaged hygroscopicity, and, if it is used vice versa together with number size distributions, will enable to derive realistic CCN number concentrations. While $\kappa$ does not reflect the full variability of the atmospheric aerosol, it does a good job in the described manner when used how and what it was designed for.

- Also, how well does the estimated K represent a multi-modal aerosol system, say having a distinct nucleation and accumulation mode as mentioned in the study for the moderate dust periods? Based on these discussions, what is the relevance of the claim in L9-10 in P17?

The sentence in question here is "Therefore, the CCN-derived hygroscopicity for dust and marine periods in the size range between 40 and 140 nm shows no significant difference. This is a surprising but also important result, as this may suggest the use of generalized values for $\kappa$ for related cases."

As stated above, an internal mixture is assumed, so indeed, different modes would not be resolved by this one $\kappa$ value at one given $d_{crit}$. However, one could have assumed that an aerosol which is predominantly related to clean marine air masses will have particles in the discussed size range (40 to 140 nm) which have, even when assumed to be internally mixed, a different hygroscopicity than those in a period characterized by large desert dust contributions, due to the predominance of particles modes from different origins in the two different cases.

However, this was not what we found. Instead, even for the vastly differing cases of marine and dust related air masses, "$\kappa$ during the marine period generally agreed with that of the dust period within uncertainty". And it should be stressed that this result is valid for the mentioned size range, which is a crucial one in which particles either do or do not act as CCN, hence determining the overall CCN concentrations. So summarizing, our results can be an important contribution to possibly simplify respective modeling efforts. Your remark here does not question our claim.

What is the 'overall average K value' (L2, P9), an average of the K values for all the supersaturations? If so, how it can represent the overall hygroscopicity of the aerosol system?

The average $\kappa$ values included all obtained values for the size between ~30 to 160 nm (critical diameter range) particles. We added the following:

"As this value was derived for different supersaturations and hence different particle sizes, it has limited use, only. However, in Sect. 3.3, we will discuss $\kappa$ values in more detail related to most extremely differing air masses. $\kappa$ values did not vary much between these air masses nor between different particle sizes, so that the here given average value may be of some use to characterize the aerosol at least roughly."

- There is confusion with the data availability. Each parameter seems to have different periods of availability. It is mentioned in the Introduction (not even in the Experiment and Methods section), and so hard to follow during the Results and Discussion. How much period does Fig. 1 represent? It will be better if the measured parameters along with their observation period are presented as a table.

Figure 1 represents data from April 2008 to December 2017.

We added a new table (Table 1 in the new version of the manuscript) to summarize all data information, including measured parameters, instrumentation, time resolution, and sampling periods.

- The absorption coefficient is corrected using a theoretically (Mie) derived scattering coefficient assuming a 'less absorbing' marine aerosol system. However, the same study highlights the seasonal presence of dust aerosols. In that case, how relevant is the scattering correction applied to the reported absorption coefficient values?

While the correction applied here is needed, as described in the text, it is not perfect. To make this clear, the authors have inserted the following sentences on page 5, line 29:

"However, due to the assumption of a refractive index and the assumption of spherical particles, the quality of calculated scattering coefficients are not sufficiently good, e.g., for the use in radiative transfer calculations. Therefore, scattering data are not presented. The calculated scattering coefficients serve as the best estimate for minimizing artifacts in the absorption measurements."

And at the end of the paragraph, one sentence further:

"It can be seen later in the study (Sect. 3.2.1.), that further improvements for the scattering corrections are of importance, especially for high single-scattering albedos since Eq. 2 is merely a first order correction. A deeper investigation requires instruments not affected by scattering artifacts, e.g. photo-acoustic photometers."

- The introduction needs a thorough revision. The authors should clearly specify the objectives and relevance of this paper systematically. Why the unsupervised Ml is preferred in this study as mentioned in L3, P3? The data strength and location details can be moved to the later (Experiment and Methods) section.

Our impression from both reviews was, that stronger reasoning for our choice of the machine learning method was needed upfront in the text. Therefore, we added more information on this topic in the introduction section to explain the difference between supervised and unsupervised machine learning (page 3, line 12).

In this study, we did not have preassigned clusters before we performed the machine learning algorithm. Therefore, unsupervised machine learning algorithms were the better solution. We

know that particles in different size ranges have different formation routes, sources, and behaviors. For example, we can assume that aerosols with high Aitken and low coarse mode are different from aerosols with low Aitken and high coarse mode. The high or low concentrations of Aitken and coarse mode particles can be reified into a distance in space. This is similar logic to the k-means clustering algorithm, which uses the Euclidean distance to measure the similarities between objects. Therefore, we used the k-means in this study and it successfully classified the aerosol particle into different types.

We wonder if the reviewer can accept the introduction as it is now. We feel that it follows a central theme which we did not want to cap unless the reviewer will still express a strong discontent.

- Another concern is the lack of appropriate references which might have enriched the discussions more. A few examples are;

  Section 2.4: studies like Furutani et al., 2008; Jayachandran et al., 2017; 2021, etc has followed this approach at other parts of the globe

  Studies like Nair et al., (2020) have investigated the CCN characteristics during the mixing of distinct air masses based on the clustering of aerosol NSD, which are not cited or discussed.

First of all, let us be clear that we looked at the correct publications by citing the doi of these publications here:

Furutani et al. (2008): https://doi.org/10.1016/j.atmosenv.2007.09.024

Jayachandran et al. (2017): https://doi.org/10.1016/j.atmosenv.2017.06.012

Jayachandran et al. (2021): https://doi.org/10.1016/j.atmosres.2021.105976

Nair et al. (2020):  https://doi.org/10.5194/acp-20-3135-2020

If these are the publications you meant, we are sorry, but we do not understand this comment.

Section 2.4. introduces the derivation of particle hygroscopicity, originally suggested by Petters & Kreidenweis (2007). Since then, this method has been used in a lot of studies by a lot of different groups, and also by us. We do not see how adding a few of these studies, e.g., the ones you suggested, or others, would improve this section. We feel that this does not belong to the method introduction.

Moreover, the studies you mention are neither connected to our study area nor to the focus of our study (examining a long-term data-set which covers several years). We did not see an easy and straightforward way to include these publications without omitting others, as our focus is not about reviewing results on CCN but on proposing machine learning for discovering patterns in long-term data.

We did, however, include Jayachandran et al. (2022) (if we found the correct study, this is actually only printed just now, so the year is 2022). We did this towards the end of Sect. 3.2.2:

"A somewhat comparable result of similar κ for differing aerosol was obtained recently by Jayachandran et al. (2022), examining aerosol during the Indian summer monsoon: Their κ values scattered over a wide range. But when comparing mean and median values for dry and wet periods they were similar, although total particle number concentrations and $N_{CCN}$ varied strongly.

- **The LAC data is mentioned and a monthly mean picture is shown. But, no more discussions on that! Any reasons?**

The light absorbing carbon (LAC) data originally was not included in the classification as

a) no LAC data was available for 17.52% of the measurement period for which particle number concentration data is available, and b) we wanted to focus on the more often used and more widely needed particle number size distributions.

However, as this topic was mentioned in both reviews, we now included and discussed the LAC data sorted according to the derived clusters in Sect. 3.2.1 and a new figure (Fig. 8 in the new version of the manuscript). The main part of this discussion is at the end of this section and is as follows:

"While, as mentioned above, $\sigma_{abs}$ and $\sigma_{abs,corr}$ were the lowest for the marine type cluster and the highest for the heavy dust cluster, the pattern showing up for these LAC data (Fig. 8) are not fully conclusive. It is striking that the moderate dust cluster has the second-lowest LAC values, although one could assume that the moderate and heavy dust clusters would be more similar in this respect. On the other hand, the freshly formed cluster, which resembles the marine cluster plus additional small freshly formed particles, has much higher LAC values than the marine cluster, while strong absorption by freshly formed particles is not to be expected. We assume that large uncertainties presented by the influence of light scattering are causing these issues. Therefore, this suggests that further measurements with methods that are not influenced by the light scattering coefficient are necessary. These can be photo-acoustic photometers for measuring the light absorption coefficient or Single Particle Soot Photometers (SP2) for measuring the refractory black carbon."

**Specific comments**

L13, P2: the 'physics, chemistry, and biology' usage seems too qualitative!

We changed this sentence and extended the information to be more precise.

"Marine aerosol particles' hygroscopicity and their ability to act as CCN can be controlled by marine ocean processes such as biological activity and wind-dependent sea spray generation (Quinn et al., 2015). For example, in the North Atlantic Ocean, O'Dowd et al. (2004) found that

the organic fraction dominated the sub-micron aerosol mass and contributed 63% (45% water-insoluble and 18% water-soluble) during algae bloom periods, while this value decreased to 15% during the lowest ocean activity periods."

L5-6, P3: What is the relevance of this statement?

As machine learning becomes more and more important for the evaluation of large data-sets in our community, this sentence should inform readers about the fact that there are machine learning algorithms which are freely available and easy-to-use in many platforms, to help encourage their use. We made a slight amendment by adding one example: ", such as Scikit-learn in Python".

Figure 2 is specified as the monthly mean, but there is no such information about Figure 1. Is it hourly mean?

The LAC and PNSD data are hourly mean values. We added a new table (Table 1 in the new version of the manuscript) to summarize this information. We also added "hourly averaged" to the caption of Fig. 1.

Fig. 7: What is the significance of linear scaling apart from the logarithmic one? Not clear from the text.

It is clear to see the Aitken and accumulation mode, and Hoppel minimum in the linear y-scale plot. The super-micron or sea-spray aerosol mode can be better seen in the logarithmic y-scale plot. We show both linear and logarithmic y-scale for the convenience of readers.

L7-11, P13: Confusing. It is obvious that the nucleation mode particles contribute less to the volume as the mass distributions. As seen in the figure, the nucleation mode aerosol system will have a large total aerosol concentration. But it is mentioned that the NPF happens in marine air mass with low particle concentrations. Please justify this statement.

To make this part easier to read, we divided it into three sentences. Also, we changed "low particle number concentrations" to " low particle volume concentrations".

As an additional explanation, we amended the preceding sentence:

"While we did not measure PNSDs below 20 nm, it can still be safely assumed that such high concentrations of small particles indicate new particle formation events which must have happened recently in the respective air masses, which is known to occur e.g., in the marine free

troposphere, followed by downward mixing of the particles (Korhonen et al., 2008, Merikanto et al., 2009)."

L28, P15: during 'this' previous study??

We changed "this" to "the". Now it is: "The freshly-formed and heavy dust clusters did not show up during the previous study and are described here for the first time for CVAO."

L7-9, P16: Not clear.

This refers to "Also, it is clear that the atmospheric $N_{HM}$ (i.e., $N_{CCN,0.30\%}$), meaning also particles being able to act as CCN, is about 2.5 times higher during dust periods than during marine periods at CVAO."

We reformulated this to:

"Also, it is clear that the atmospheric $N_{HM}$, which best represents number concentrations to be expected for CCN in the atmosphere, is about 2.5 times higher during dust periods than during marine periods at CVAO."